# Characterization and Corrosion Behavior of Zinc Coatings for Two Anti-Corrosive Protections: A Detailed Study

Alina Bianca Pop [1,*] , Gheorghe Iepure [2] , Aurel Mihail Titu [3,*] and Sandor Ravai-Nagy [1,4]

1   Department of Engineering and Technology Management, Faculty of Engineering,
    Northern University Centre of Baia Mare, Technical University of Cluj-Napoca, 62A, Victor Babes Street,
    430083 Baia Mare, Romania; sandor.ravai@imtech.utcluj.ro
2   Department of Mineral Resources, Materials and Environmental Engineering, Faculty of Engineering,
    Northern University Centre of Baia Mare, Technical University of Cluj-Napoca, 62A, Victor Babes Street,
    430083 Baia Mare, Romania; gheorghe.iepure@irmmm.utcluj.ro
3   Industrial Engineering and Management Department, Faculty of Engineering,
    "Lucian Blaga" University of Sibiu, 10 Victoriei Street, 550024 Sibiu, Romania
4   Institute of Technical and Agricultural Sciences, University of Nyíregyháza, 31/b, Sóstói Street,
    4400 Nyíregyháza, Hungary
*   Correspondence: bianca.bontiu@gmail.com (A.B.P.); mihail.titu@ulbsibiu.ro (A.M.T.)

**Abstract:** The purpose of this research is to characterize and evaluate the corrosion behavior of zinc coatings used for corrosion protection, with a special focus on the S235 steel material. The introduction highlights the need for corrosion protection in industrial settings, as well as the importance of understanding corrosion processes and the development of corrosion products to develop more effective solutions. The study's goals are to undertake an extensive analysis of corrosion products formed on the zinc coating's surface, to evaluate the performance of these coatings under atmospheric circumstances, and to investigate the effect of deposition parameters on coating quality. The essential message provided to readers is the critical significance of knowing corrosion product formation mechanisms and zinc coating corrosion behavior in developing long-lasting and effective protection measures. The study methodology includes cycle testing, morphological and chemical examination of corrosion products, as well as optical and electron microscopy and energy-dispersive spectroscopy. Corrosion resistance is assessed using accurate measurements. The results show that zinc coatings have exceptional corrosion resistance under air settings, with the produced corrosion products offering further protection to the underlying material. Furthermore, the study demonstrates that the surface roughness of S235 steel has a substantial impact on the quality and corrosion behavior of hot-dip galvanized coatings. The findings emphasize the necessity of detailed characterization of corrosion products, the effect of depositional factors on zinc coating performance, and the need for novel corrosion protection methods. These discoveries have significant implications for the corrosion protection sector, providing the potential to improve the longevity and efficiency of protective systems used in industrial applications.

**Keywords:** corrosion protection; zinc coatings; S235 steel; corrosion behavior; corrosion products; deposition parameters; performance evaluation; surface roughness; durability; industrial applications

## 1. Introduction

Corrosion prevention is critical for maintaining the integrity and lifespan of materials exposed to corrosive conditions. Zinc coatings have attracted significant attention among other protective coatings due to their low cost and demonstrated corrosion resistance. Understanding the behavior and features of zinc coatings is critical for creating successful corrosion prevention and material durability methods.

The need for this study stems from the continual need to develop corrosion prevention technology and fill information gaps. Despite substantial study in the discipline, numerous

critical areas deserve more exploration. To begin, a thorough understanding of the development and behavior of corrosion products on zinc coatings is required to improve their protective characteristics. Second, the impact of deposition parameters such as current density and surface preparation on the quality and performance of zinc coatings is still a major study area. Finally, investigating the effect of surface roughness on the corrosion behavior of zinc-coated materials is crucial for designing customized corrosion protection systems.

The originality and novelty of this study stem from its complete approach to characterizing and evaluating zinc coatings, with a special focus on S235 steel material. This work seeks to give significant insights into the mechanisms behind corrosion processes, the impact of deposition parameters, and the influence of surface roughness on coating performance by integrating morphological, chemical, and corrosion testing approaches. Such a comprehensive examination is critical for creating new corrosion prevention techniques and closing knowledge gaps.

The primary research goals of this study are twofold: (1) to characterize the formation and behavior of corrosion products on zinc coatings under atmospheric conditions, and (2) to assess the influence of deposition parameters and surface roughness on the quality and corrosion resistance of zinc coated S235 steel. Achieving these objectives will help to design more efficient and long-lasting corrosion prevention technologies.

This paper is organized as follows: Section 1 discusses the significance of corrosion prevention and identifies research needs on the subject. Section 2 summarizes the current state of the research. Section 3 discusses the study's materials and methodologies, including sample preparation, characterization techniques, and corrosion testing processes. Section 4 covers the investigation's findings, which include the characterization of corrosion products, the evaluation of coating performance, and the impact of deposition parameters. Section 5 analyzes the ramifications of the results and gives insights into the research's practical applicability. Finally, Section 6 summarizes the key findings, emphasizes their relevance, and suggests future research areas.

This study intends to contribute to the improvement of corrosion protection technologies and give significant insights for scholars and practitioners in the area by addressing these research objectives and giving a detailed analysis.

## 2. Current State of Research

Galvanized steel, with its excellent corrosion resistance and widespread use in various industries, remains a topic of extensive research and development. Numerous studies have focused on investigating the behavior, performance, and optimization of zinc coatings for enhanced corrosion protection. This section provides an elaborate overview of the current state of research in the field.

Galvanized steel is utilized in many applications that require excellent corrosion resistance. Mechanical characteristics of the deposited layer are critical and are determined by the microstructure of the interface between the substrate and the coating [1,2]. The size of the steel substrate influences the kinetics of Fe-Zn phase nucleation during immersion in the zinc solution during the zinc plating process. In a pure Zn solution, the grain size of the steel substrate has no influence on the development of gamma, delta, and zeta phases; however, the presence of at least 0.2% Al causes the creation of a Fe-Al (Fe2Al5) inhibitory layer and the slowdown of Fe-Zn phase nucleation [1]. When the retention period in molten zinc is extended, the inhibition layer above the large-grained substrate (85 m) at the grain boundary area dissolves by zinc, resulting in the formation of Fe-Zn phases [1].

Petit et al. [3] performed a thorough investigation of the influence of steel substrate size on the microstructure and corrosion behavior of hot-dip galvanized coatings [1,3]. Their findings provide light on the dynamics of Fe-Zn phase nucleation and the creation of inhibitory layers during the zinc plating process.

Bolzoni et al. [4] emphasized the corrosion performance of galvanized steel in varied conditions, highlighting the need of knowing unique corrosion processes and zinc coating

protective characteristics. This study emphasized the importance of additional research on the topic.

Le et al. evaluated the influence of deposition parameters, such as current density and surface preparation, on the microstructure and corrosion resistance of zinc coatings [5]. Their findings emphasized the significance of these parameters in improving coating quality and performance.

Morcillo et al. investigated the effect of surface roughness on the corrosion behavior of zinc-coated steel in marine environments [6]. Their study discovered a relationship between surface roughness and corrosion start and propagation, providing significant information for corrosion control methods.

Li et al. focused on the generation and characterization of corrosion products on zinc coatings, providing an in-depth understanding of their role in corrosion protection [7]. Their research aided in the improvement of coating design and performance.

Galedari et al. investigated the corrosion resistance and barrier characteristics of zinc-rich thermal spraying coatings [8]. Their research looked at the potential of these coatings for long-term corrosion prevention in harsh settings.

Guo et al. evaluated the influence of post-treatment techniques on the corrosion resistance of zinc coatings, such as chromate conversion coating and silane-based coatings [9]. Their findings provided insight into post-galvanizing procedures for increasing the durability of zinc-coated surfaces.

Pokorny et al. investigated galvanized steel corrosion behavior in alternating wet-dry circumstances, simulating real-world air exposure [10]. Their research assessed degradation mechanisms and made recommendations for enhancing zinc coating compositions for prolonged service life.

Abd El-Lateef et al. investigated the self-healing behavior of zinc coatings using corrosion inhibitors [11]. Their study sought to develop self-healing technologies capable of mitigating the impacts of coating deterioration and extending the lifespan of zinc-coated structures.

Yan et al. looked at the effect of alloying elements like aluminum and magnesium on the microstructure and corrosion resistance of hot-dip galvanized coatings [12]. Their research illustrated how alloying may be used to adjust the characteristics of zinc coatings for certain purposes.

In summary, current research on galvanized steel and zinc coatings includes a wide range of investigations, such as the effects of steel substrate size, deposition parameters, surface roughness, post-treatment methods, corrosion behavior in various environments, self-healing mechanisms, and alloying elements. These investigations give a thorough understanding of zinc coating performance, corrosion processes, and solutions for enhancing corrosion protection. The current study seeks to increase roughness and knowledge by defining corrosion products, assessing the effect of deposition parameters and surface roughness, and furthering our understanding of zinc coating behavior on S235 steel.

## 3. Materials and Methods

### 3.1. Material and Equipment Used

The research material was low-carbon steel (S235J0) with the chemical composition listed in Table 1. Because of its low cost and good (high) mechanical qualities, this material is widely employed in a variety of industrial areas.

**Table 1.** Chemical composition of the S235J0 material according to the supplier specifications.

| C | Mn | P | S | N | Si | P |
|---|---|---|---|---|---|---|
| 0.17 | 1.40 | ≤0.035 (0.050) | ≤0.035 (0.050) | ≤0.12 | - | (0.050) |

S235J0 steel is easily treated using typical machining methods.

Turning was used to retrieve test specimens from annealed rolled steel. The specimens for analysis have a cylindrical shape with the following dimensions: diameter $\phi$ = 22 mm and length l = 25 mm.

Three batches of 5 specimens each were prepared, corresponding to the three galvanizing methods. Each batch was numbered from 1 to 5, with specimens sharing the same number having the same surface roughness. The specimens were coded according to the galvanizing process used, namely electro-galvanizing—GE1...GE5, hot-dip galvanizing—HDG1...HDG5, and hot-dip and spin galvanizing—HDGC1...HDGC5.

The specimens were machined at the specified roughness values in Table 2 using a CNC lathe, HAAS TL-2. Turning was performed with a carbide cutting tool: VBMT110308-FP6, WPP20S (Walter Tiger tec, Walter AG, Tübingen, Germany). This type of carbide is commonly used for medium-level machining and finishing operations on steel materials. Its multilayer coating provides high wear resistance, which is necessary for processing metal materials.

**Table 2.** The roughness of the specimens after turning and coating.

| Section | 1 | 2 | 3 | 4 | 5 |
|---|---|---|---|---|---|
| | Roughness Ra [μm] | | | | |
| Roughness Substrate | 0.684 | 1.512 | 2.488 | 3.635 | 5.721 |
| Roughness after Coating—HDG | 1.379 | 2.051 | 2.755 | 3.848 | 5.425 |
| Roughness after Coating—HDGC | 1.907 | 2.350 | 2.663 | 3.541 | 5.213 |
| Roughness after Coating—EG | 0.435 | 1.238 | 1.913 | 3.673 | 6.323 |

The data in Table 2 are extracted from the authors' paper [13] and represent a summary of the evolution of surface roughness after different galvanizations. The study was oriented to determine the influences on the surface quality of the specimen after galvanization depending on the type of galvanization and the roughness of the initial substrate. In the present study, the specimens are also studied from the metallographic point of view of the zinc layers deposited with different technologies as well as their corrosion resistance.

The sample's surfaces were machined to various roughness levels. Table 2 shows the average roughness values of the turned specimens. After turning, pickling, and galvanizing, roughness measurements were taken in the longitudinal direction (along the generators) with a Mitutoyo SURFTEST SJ-210 roughness tester (Mitutoyo, Kanagawa, Japan).

Because the cutting tool used in the turning process corresponds with this orientation, longitudinal roughness measures were used. The profile, thickness, and shape of the zinc coating, as well as the compounds originating from zinc contact (diffusion) with the surface layer of the steel specimen, may therefore be evaluated by performing a cross-sectional examination of the specimens.

Although measuring roughness in both the longitudinal and transverse directions would be ideal, the choice was taken to measure in the longitudinal direction to assure realistic findings that are relevant to the steel's performance in diverse applications. Because the specimens were rotationally symmetric, it was possible to measure roughness just along the cylinder generator. As a result, the longitudinal roughness measured after cylinder creation is usually greater than the transverse roughness recorded after cylinder generation.

### 3.2. Galvanizing

The most important application of zinc is as a corrosion-resistant coating for steel. Electroplating or thermal hot-dip galvanizing, which is the most extensively used and appropriate zinc coating process, can be utilized to achieve zinc coating.

The zinc coating procedure was carried out in the following ways.

Electroplating of the specimens (EG) was performed as follows: degreasing and rinsing in an alkaline solution (AK16 + RV111 compounds; T = 60 °C; t = 10 min), water rinsing (T = 20 °C; t = 0.5 min), pickling (HCl + BEF30 solution; T = 25 °C; t = 5 min), electro-

degreasing and anodic cleaning (E1-DEG solution; T = 25 °C; t = 4 min), electroplating with zinc (Slotonit OT solution; T = 20–25 °C; pH: 5–5.5; t = 30 min; I = 1.5–2.0 A, U = 100 V), passivation (Slatopas Z20Blue + HNO3 solution; T = 20–25 °C; t = 1 min; pH: 1.7–2), and drying (T = 50 °C; t = 10 min).

Hot-dip galvanizing (HDG) and hot-dip galvanizing with centrifugation (HDGC) were performed as follows: degreasing and rinsing in an alkaline solution (Ferro clean 7135/1; T = 50 °C; pH: 6.1; t = 15 min), water rinsing (t = 10 min), pickling (HCl 29%; T = 25 °C; t = 20 min), water rinsing (t = 10 min), fluxing (T = 45 °C; pH: 4.0; t = 5 min; ZnCl2), drying (T = 80 °C; t = 40 min), immersion in molten zinc (T = 550 °C for HDG and T = 600 °C for HDGC; t = 5 min), cooling, and passivation (t = 1–2 min).

In the case of hot-dip galvanizing, when the steel is immersed in molten zinc, a series of intermetallic alloy layers of Fe-Zn are formed, which are metallurgically bonded to the base metal and exhibit superior properties compared to iron. These layers can consist of the following phases according to the equilibrium diagram of the Fe-Zn system: $\gamma$ phase (Fe5Zn21; 21–28% Zn; BCC, $\rho$ = 7.36 g/cm$^3$), $\eta$ phase (FeZn10; 7–11.5% Zn; HCP; $\rho$ = 7.24 g/cm$^3$), $\delta$ phase (FeZn13; 5.7–11.5% Zn; monoclinic; $\rho$ = 18 g/cm$^3$), and $\zeta$ phase (Zn, maximum 0.003% Fe; HCP; $\rho$ = 7.14 g/cm$^3$).

The roughness of the specimens after zinc coating varied depending on the galvanizing method used. The average roughness values of the coated specimens are presented in Table 3.

**Table 3.** The roughness of the specimens after zinc coating.

| $R_a$ (µm) | 1 | 2 | 3 | 4 | 5 |
|---|---|---|---|---|---|
| $R_a$ Initial | 0.684 | 1.512 | 2.488 | 3.635 | 5.721 |
| GE | 0.669 | 1.569 | 2.388 | 3.591 | 6.005 |
| HDG | 0.571 | 1.511 | 2.632 | 3.649 | 5.893 |
| HDGC | 0.722 | 1.435 | 2.562 | 3.740 | 5.425 |

The salt spray corrosion test was used to assess the corrosion resistance of the specimens. The test followed the ISO 9227:2017 standard (corrosion testing in artificial atmospheres—salt spray tests). A salt spray chamber (salt spray cabinet) SF/450/CASS produced by C&W Specialist Equipment in Mississauga, ON, Canada was utilized. The following test settings were used: salt—NaCl; T = 35 °C; salt solution concentration: 50 g/L; and salt solution flow rate: 1.5 mL/h.

Surface examination of the specimens was performed using optical and scanning electron microscopy (SEM) (Hitachi, Tokyo, Japan). A Kruss stereoscopic microscope with white light (LED) (ZEISS, Jena, Germany) was employed for surface examination. A Neophot 21 microscope—Carls Zeiss Jena (ZEISS, Jena, Germany), lit with halogen light, was used for microstructure investigation of the specimens and the deposited coating (in cross-section). A Hitachi SEM (Hitachi, Tokyo, Japan) was used for the electron microscopy examination (HITACHI SU-1510 Scanning Electron Microscope, resolution: 3.0 nm, sample stage: axis X: 0 to 80 mm; axis Y: 0 to 40 mm; axis Z: 5 to 50 mm; rotation by axis *X* and *Y*: −20° to 90°; rotation by axis *Z*: 360°, accelerating voltage: up to 15 kV).

The specimens were longitudinally sectioned before being ground with abrasive paper and polished with alumina powder on a cloth for microscopic inspection.

### 3.3. Surface Morphology of the Zinc Layer

The steel specimens were longitudinally sectioned for microstructure examination, and a metallographic sample was created by grinding, polishing, and etching with a reagent (3% nitric acid in ethyl alcohol). Due to the low carbon content, the structure is a ferrite–pearlite structure, as illustrated in Figure 1, with polyhedral grains, because the material of the specimens was annealed.

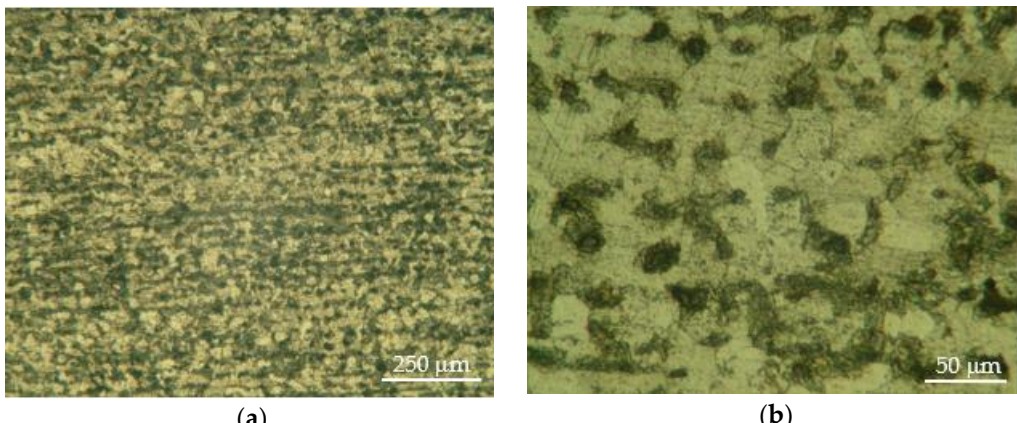

(**a**)          (**b**)

**Figure 1.** The microstructure of S235J0 steel; (**a**) microstructure of the ferritic–pearlitic substrate; (**b**) microstructure detail with distinction of the outline of the phases and constituents.

A stereoscopic microscope (ZEISS, Jena, Germany) was used to examine the specimens' surfaces. The surfaces of the specimens with the lowest and maximum roughness were examined, and macroscopic pictures of the surfaces were obtained using a stereoscopic microscope, as shown in Figures 1–3. These photos show the surface properties of both the machined and galvanized specimens in detail. The surface quality (roughness) of the specimen may be seen in the photos (Figure 2). The electrogalvanized surface has a bright metallic look and shows minimal variation when compared to the roughness of the substrate surface, whether it has low roughness (Figure 2a) or high roughness (Figure 2b). Electrogalvanization produces a pure zinc layer, which is generally a single-phase structure. This method is commonly used for components where the thickness of the coating should not exceed certain limits, typically ranging from 4–15 μm (30 to 100 g/m$^2$), and in some cases, even thinner coatings of up to 1.5 μm (10 g/m$^2$) are applied.

In the case of electro galvanically deposited zinc coating, at high current densities of 250 mA/cm$^2$, X-ray diffraction analysis revealed a weak presence of Fe on the surface of the electrodeposited layer. However, at low current densities (50 mA/cm$^2$), a higher quantity of iron atoms became noticeable.

Hydrogen evolution occurs during the electrolytic deposition of zinc [14]. At a steady rate of electrodeposition, the number of gas blisters grows. The rates of development of gas blisters and coating thickness might cause a self-oscillating regime, resulting in the production of a stratified distribution of gas pores (hydrogen carriers) [14]. When heated, the atomic hydrogen in the pores diffuses largely along the coating borders and changes into molecular hydrogen, forming confined blisters or open craters on the coating surface [14].

The roughness of the coating layer determines its quality. If the substrate surface roughness rises greatly (rapidly), dendritic development becomes non-uniform, and the average dendrite length increases [15]. This explains the crystal coarsening on the zinc coating surface caused by the presence of a sudden roughness variation (protrusion) in a small area of the substrate surface (Figure 3), where zinc crystal nucleation and growth occur in multiple directions (Figure 3) as opposed to the basal direction [15–17]. The form and size of the dendrites are also influenced by the current density. Longer dendrites form with a considerable size dispersion as current density increases, as seen on the margins of electrogalvanized steel strips where the roughness is considerably higher than on the flat surface [15].

The orientation of single-crystal or polycrystalline grains can explain the color variance on the coated surface. Polycrystalline grains feature numerous origination domains and appear to have developed from a single nucleation point [18]. Single crystals of zinc nucleate preferentially in the (0001) direction, but polycrystalline grains have orientation domains related to crystallographic planes such as $\langle 10\overline{1}0 \rangle$, $\langle 11\overline{2}0 \rangle$, and $\langle 0001 \rangle$. The zinc growth mechanism prefers grains with parallel basal planes to the substrate plane [18].

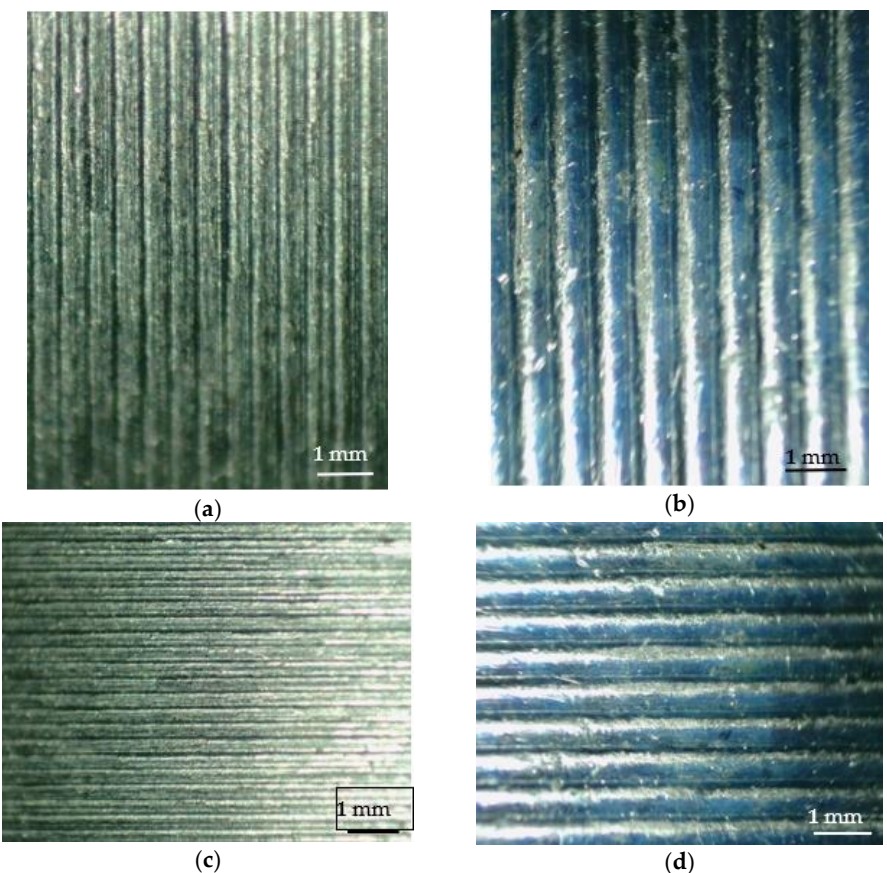

**Figure 2.** Images of the surface of the GE specimens: (**a**) finely turned (GE1); (**b**) roughly turned (GE5); (**c**) GE fine (1–4.5×, Figure 1); and (**d**) roughly (5–4.5×, Figure 3) stereoscopic images.

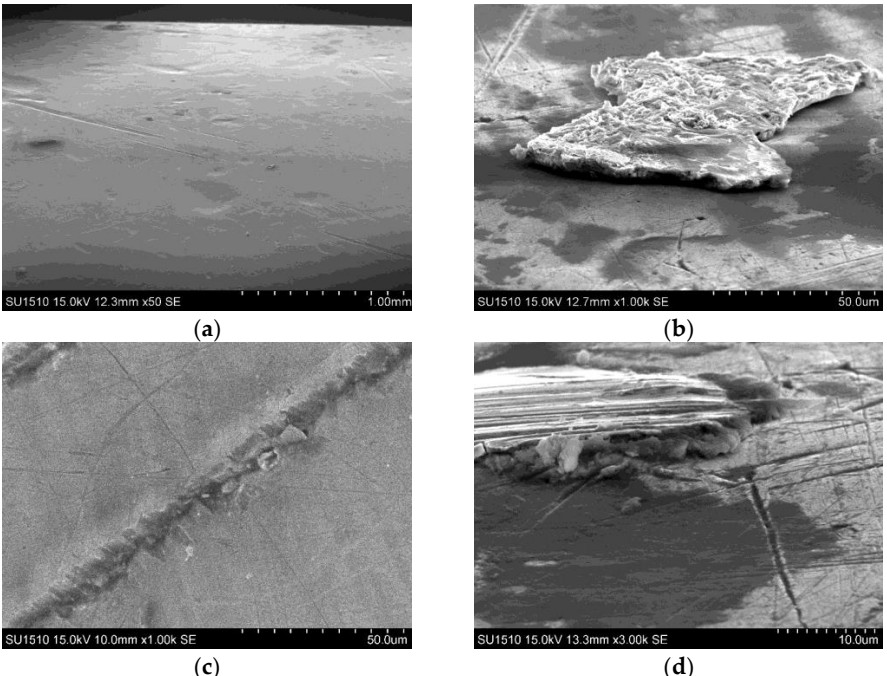

**Figure 3.** Crystal growth of zinc and zinc oxide film on the surface of GE1 parts with low roughness. (**a**) micro-craters, (**b**) conjoined prominences, (**c**) conjoined prominences, (**d**) intergranular indentations.

SEM examination revealed minor flaws such as micro-craters (Figure 3a), conjoined prominences (Figure 3b,d), and intergranular indentations (grooves) (Figure 3d) on the surface of electrogalvanized samples (GE1). These flaws on the machined surface are like those found on galvanized sheets. Individual crystal formation may be seen in the striations and surrounding the prominences, which are commonly Fe-Zn intermetallic complexes with the phase, FeZn13 [19].

Surface imperfections degrade the quality of galvanized items. The discovered flaws in galvanized steel sheets generated in a continuous galvanizing process include scratches, holes (micro-craters), prominences, and wrinkle bands, all of which impact the corrosion behavior of the coated sheet [20]. Corrosion behavior (salt spray test and polarization test) revealed that the severity of the defects affected corrosion resistance; the presence of voids reduced the overall corrosion resistance of the galvanized sheet by approximately 40%, and prominences by 10% when compared to samples without defects [20].

## 4. Results and Discussion

ImagePro Plus software (Media Cybernetics, Rockville, MD, USA) (version 7.0) was used to analyze surface defects in terms of their quantity (quantification), shape, and area occupied. The total surface area evaluated for all samples was 1100 μm × 770 μm.

As a result, the results of the analysis on the electrogalvanized surface are shown in Figure 4. The total analyzed surface area was 847,000 $\mu m^2$, equivalent to 0.847 $mm^2$. Surface defects were counted (36 samples) using software (Figure 4a) and classified into classes based on their maximum diameter (represented in different colors) due to the large variations in their length (striae—grooves versus prominences). From a statistical analysis of the area, the surface area occupied by these defects was 17,793.39 $\mu m^2$, which represents 2.1% of the total analyzed surface area of 847,000 $\mu m^2$ (Figure 4b). The distribution of defects according to their maximum diameter, their average diameter in relation to the surface area, and the correlation between these parameters are presented in Figure 4c.

According to the histogram analysis, most faults have maximum diameter values of less than 4 m, and the computed average diameter for the majority is less than 3.4 μm. The surface area of most defects is below 10 μm (Figure 3c).

The coating profile closely followed the substrate surface in the GE5 samples with considerable substrate roughness. The most noticeable flaws were found in places with increased substrate roughness (GE5) along the turning tool's direction, as illustrated in Figure 5.

Similarly, a series of faults were discovered on the surface of the deposited layer in GE samples with greater steel substrate roughness (Figure 5), like the defects reported in samples with smaller roughness. In this case, however, tiny protrusions (Figure 5a) were also noticed, which might be gas blisters. This is feasible because hydrogen emission occurs during the electrolytic deposition of zinc [14]. At a steady rate of electro-deposition, the number of gas blisters (swellings) rises. The rates of development of gas blisters and coating thickness might cause a self-oscillating regime, resulting in the production of a stratified distribution of gas pores (hydrogen carriers) [14]. When heated, the atomic hydrogen in the pores diffuses largely along the borders of the diffuse coating layers and converts into molecular hydrogen, resulting in the production of closed blisters or open craters on the coating's surface [14].

Surface defect analysis was carried out on a galvanized sample with a high steel substrate roughness (GE5), and the investigated surface area in this case was 1100 μm × 770 μm (847,000 $\mu m^2$ = 0.847 $mm^2$).

The analytical findings are shown in Figure 6. Surface flaws were counted (81 samples), and it was discovered that there were more faults with smaller diameters than in the GE1 sample. These faults were mainly equiaxed in form, and their total surface area was 3079.11 $\mu m^2$, representing 0.36% of the total analyzed surface area of 847,000 $\mu m^2$ (Figure 6b). Figure 6c depicts the distribution of defects based on maximum and average diameter relative to the surface area, as well as the association between these sizes. In

this situation, the maximum diameter of 50 faults (objects) was less than 1.7 μm, and the average diameter of most defects was below 1.6 μm (Figure 6c). The surface area of most defects was below 2 μm².

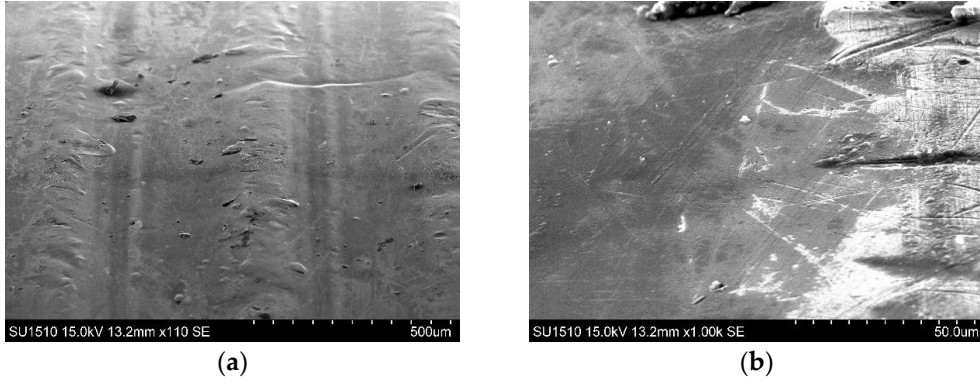

| Stats | Area | Diameter (Max) | Diameter (Mean) |
|---|---|---|---|
| | μm² | μm | μm |
| Min | 8.264463 | 4.11162 | 3.393345 |
| (Obj.#) | 30 | 33 | 40 |
| Max | 3233.058 | 473.406 | 197.0514 |
| (Obj.#) | 36 | 36 | 36 |
| Range | 3224.794 | 469.2943 | 193.6581 |
| Mean | 494.2608 | 51.92693 | 26.29727 |
| Std.Dev | 817.3143 | 87.26173 | 35.84959 |
| Sum | 17793.39 | 1869.369 | 946.7016 |
| Sample | 36 | 36 | 36 |

(**a**)     (**b**)     (**c**)

**Figure 4.** Surface analysis of GE1 for surface defect quantification: (**a**) defect numbering; (**b**) measured values; (**c**) histograms of the defect diameter to defect area ratio.

**Figure 5.** Crystal concretions of zinc and zinc oxide film on the surface of the GE5 parts with high roughness.

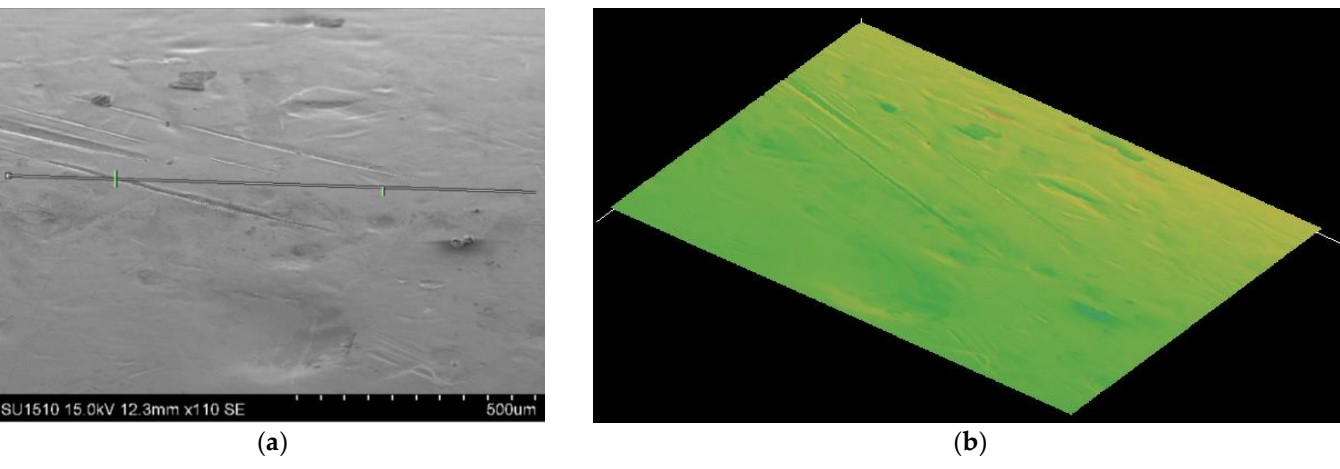

| Stats | Area µm² | Diameter (Max) µm | Diameter (Mean) µm |
|---|---|---|---|
| Min | 2.043698 | 1.6677 | 1.617572 |
| (Obj.#) | 7 | 14 | 7 |
| Max | 471.5369 | 90.52135 | 45.33001 |
| (Obj.#) | 5 | 5 | 5 |
| Range | 469.4932 | 88.85365 | 43.71244 |
| Mean | 45.95686 | 9.615728 | 6.77736 |
| Std.Dev | 68.51028 | 11.84044 | 6.069565 |
| Sum | 3079.11 | 644.2538 | 454.0831 |
| Sample | 67 | 67 | 67 |

**Figure 6.** Surface analysis of GE1 for quantifying surface defects: (**a**) defect numbering; (**b**) measured values; (**c**) histograms of defect diameter/surface area ratio.

Figures 7 and 8 show surface analysis using ImagePro Plus software to examine the profile in a specified direction and topography.

**Figure 7.** *Cont.*

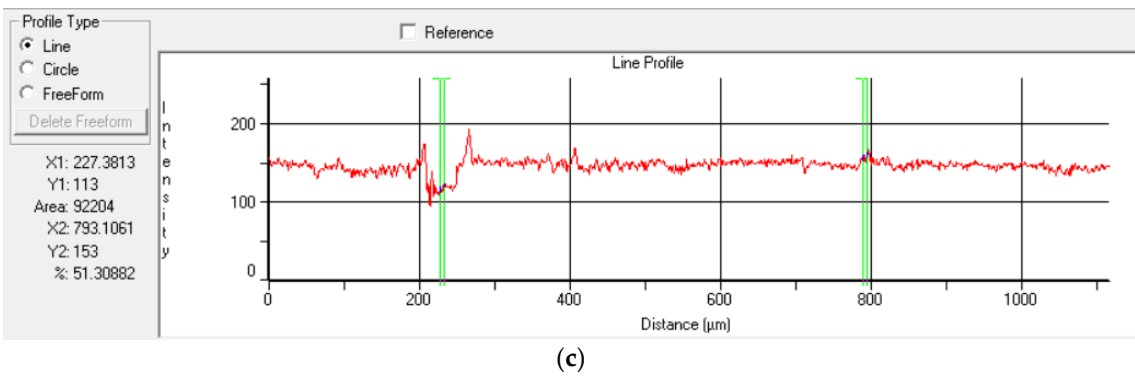

(**c**)

**Figure 7.** Topography and line profile—sample GE1 (**a**) SEM image; (**b**) topography; (**c**) line profile.

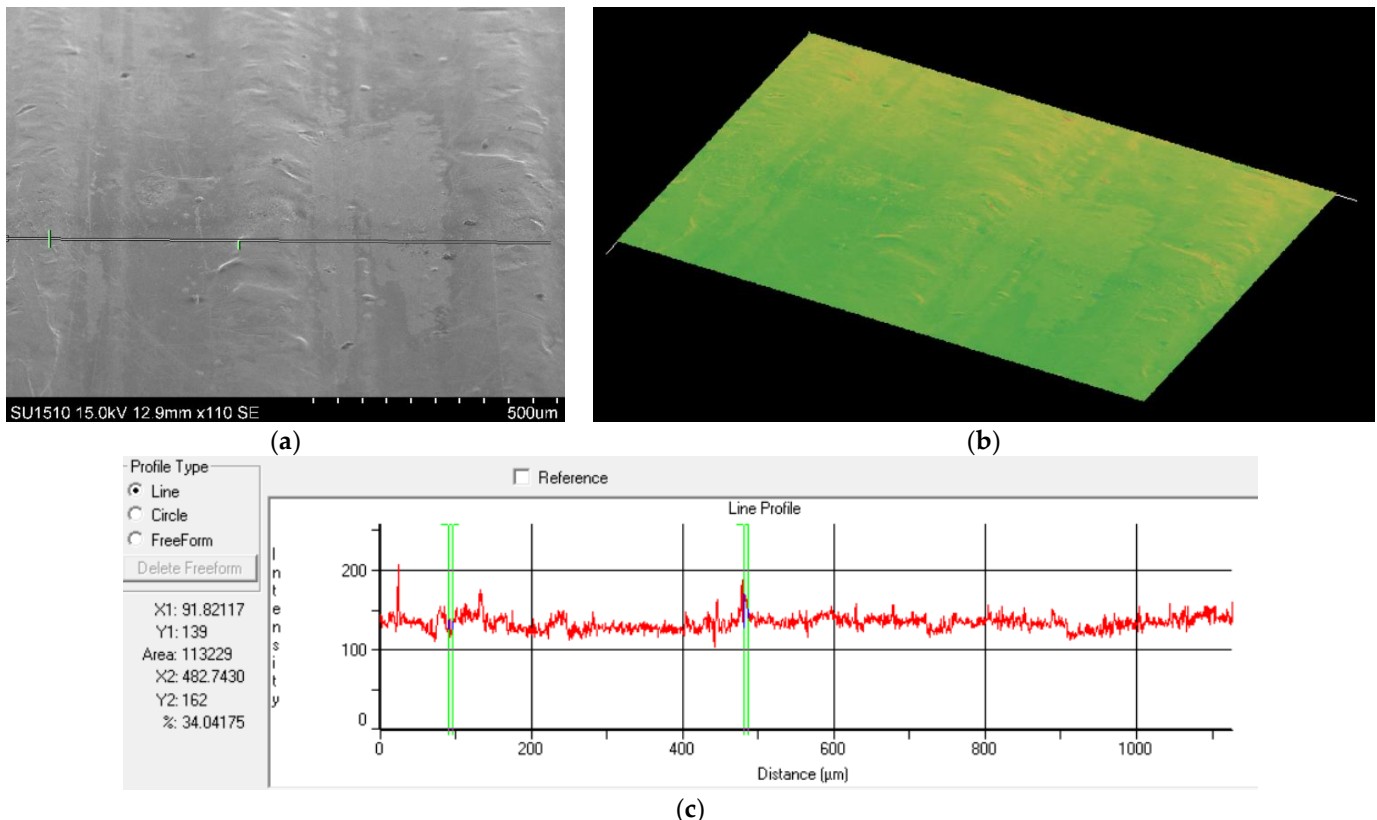

(**c**)

**Figure 8.** Topography and line profile—sample GE5 (**a**) SEM image; (**b**) topography; (**c**) line profile.

The study of SEM images and topography profiles of samples GE1 and GE2 (Figures 7 and 8) shows that the difference in substrate roughness is preserved even after zinc coating. Even after zinc coating, the sample with the greater substrate roughness (GE5) had a higher roughness. The surface examination of samples GE1 and GE5 (Figures 7 and 8) indicates that the roughness of the steel substrate affects the quality of the electrolytically formed layer.

Solidification (crystallization) happens in the hot-dip galvanizing process via nucleation and growth processes on the steel substrate, resulting in dendritic crystal formation known as spangles or flakes. Dendrites are almost two-dimensional and can range in size from millimeters to centimeters, depending on parameters such as zinc bath composition, substrate type, bath temperature, and holding time [21–23].

Dendrites (spangles, flakes) can be identified visually by their different crystallographic orientation, with darker-colored dendrites having a pyramidal orientation compared to glossy crystals [22]. Surface roughness is also affected by crystallographic orienta-

tion. Dendrites (η phase = hexagonal Zn) can grow to lengths of several centimeters [23]. The grains preferentially develop in the solidification direction from the substrate to the surface, and the grain boundaries of the phase are closed and difficult to observe at low magnifications [23].

During solidification, the zinc grains can develop in a variety of locations relative to the steel substrate [16]. The look of the zinc coating surface varies depending on the angle, the angle between the basal plane (0001), and the steel sheet surface, from shining and glossy (dominated by basal planes parallel to the substrate surface) to matte. The basal planes have the highest atom–atom bonding energy. Breaking and dissolving atomic bonds in the basal planes takes more energy than in other planes [16]. Slowly cooled samples have bigger zinc dendrites and fewer in quantity than rapidly cooled samples, which have a higher number of primary zinc dendrites due to faster nucleation rates [24].

The zinc-coated layer deposited during immersion in hot-dip galvanizing (Figure 9) is metallurgically bound (at the atomic level) and integrated with the steel, making it very adhesive. Surface roughness is noticeably different from electrogalvanizing since the surface is not as smooth and does not perfectly follow the profile of the machined specimen. This is due to the creation of an adhering protective layer of zinc and Fe-Zn complexes, which has numerous layers and undergoes varying rates of nucleation and growth [1].

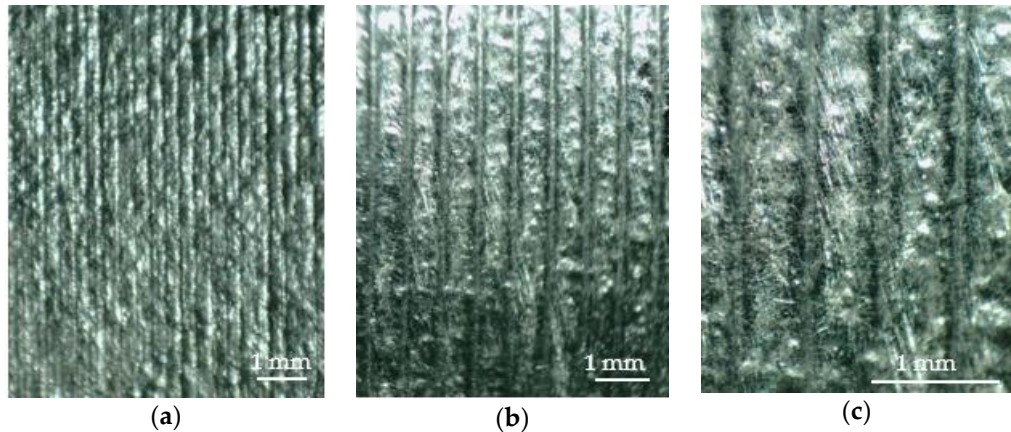

(**a**)    (**b**)    (**c**)

**Figure 9.** Surface images of HDG specimens: (**a**) finely turned (HDG1), 40× magnification; (**b**) roughly turned (HDG5), 40× magnification; (**c**) roughly turned (HDG5), 80× magnification.

The orientation of the surface grains in the coating layer is more ordered when the surface roughness of the steel substrate is low [2]. When the metallic bath includes more than 0.2% Al, a Fe-Al interfacial layer forms during the first stage of immersion. The creation and expansion of the Fe-Zn phases are significantly affected by the shape of the interfacial layer during this stage [2].

Like immersion galvanization, the deposited coating is metallurgically linked and integrated with the steel in the case of hot-dip galvanization and centrifugation (Figure 10). When compared to the immersed specimens, the surface roughness changes when the excess non-solidified zinc on the surface is drained by centrifugal force, resulting in a matte appearance due to the produced roughness.

The coating surface has a glossy look that alternates with a matte appearance, both on the turned edges and their bases, and follows the profile form of the specimen produced by the turning process. The glossy galvanized surface is the result of dendritic branch growth along the free surface, whereas the matte regions (with pits) are the consequence of primary dendrite arms growing along the steel substrate and the free surface. For the grains to have large diameters and develop along the basal plane, it is expected that the contact angles are about the same for the free surface and the steel substrate and should vary between 17 and 80 degrees relative to the surface in the case of steel sheet [25]. Other studies, however, indicate that the contact angle is equal to or greater than 90 degrees [26]. When the contact angle is equal to 90 degrees, the existence of a boundary changes the preferred growth

paths associated with interfacial energy anisotropy, and the growth rate along a particular boundary reduces as the incidence angle (or tilt angle) increases.

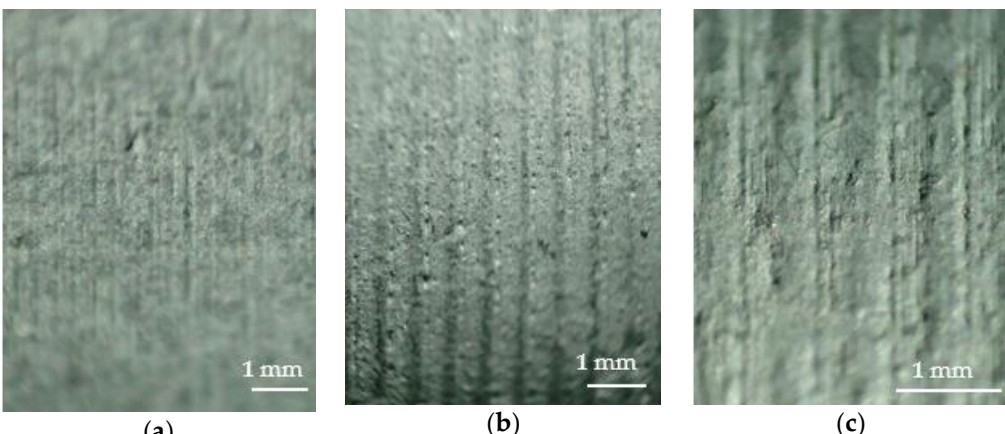

**Figure 10.** Surface images of HDGC specimens: (**a**) finely turned (HDGC1), 40× magnification; (**b**) roughly turned (HDGC5), 40× magnification; (**c**) roughly turned (HDGC5), 80× magnification.

The SEM investigation of the thermally galvanized surface (Figure 11) offers detailed information about the flaws in the deposited layer, which may also be seen by optical microscopy. Because of the distinct nucleation mode of the Fe-Zn intermetallic phases and the crystal growth mode, which may be impacted by impurities present in the metallic bath, the surface is significantly rougher than the electrolytically formed layer [3]. The number of faults and the irregularity of the surface give the specimens a matte look, as opposed to the shining appearance of the electrolytically coated specimens.

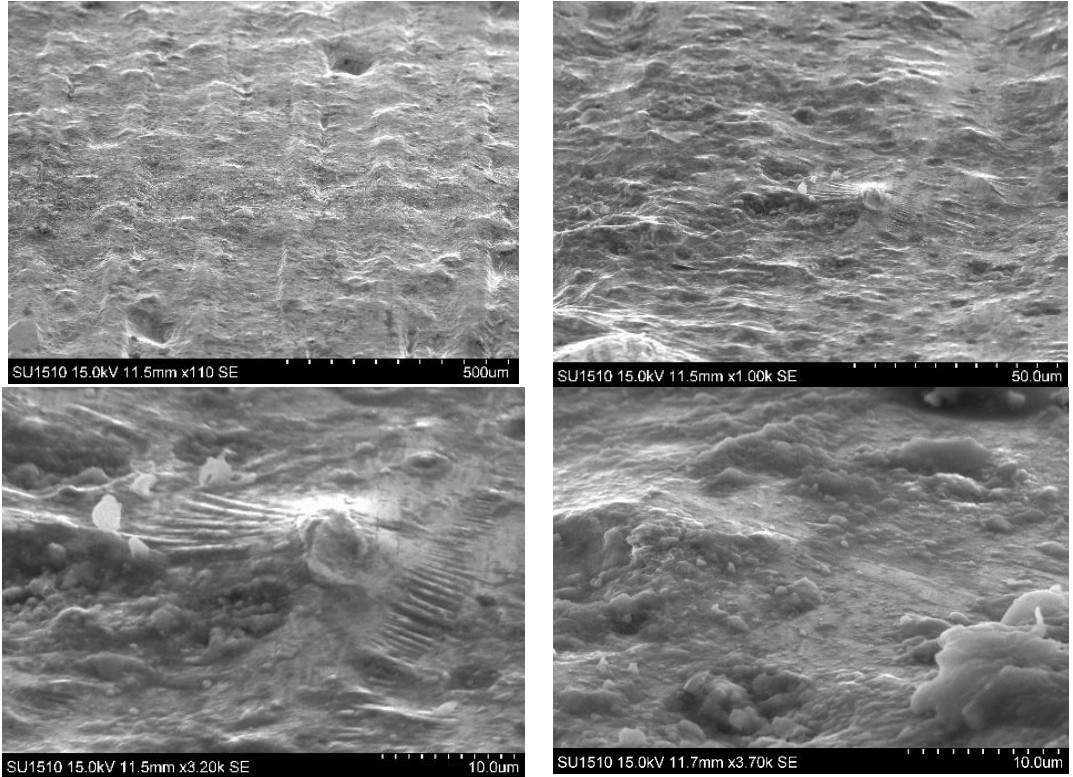

**Figure 11.** HDG 5 SEM image of sample surface.

Similarly, on the surface of the electrolytically zinc-coated samples, the same sorts of flaws are detected. Because of the impact of nucleation and the production of Fe-Zn

intermetallic phases, the coating surface deviates less from the steel substrate profile. There are cavities (craters), wrinkles, and protrusions, and the roughness is substantially higher than in the electrolytically zinc-coated samples. In addition, zinc oxide crystals were found on the surface.

The term "smooth" refers to a smooth, uniform surface, whereas "concavo–convex" refers to an uneven surface. Because a concavo–convex surface is more prone to corrosion than a smooth one, the increase in surface roughness influences corrosion resistance [16].

A sequence of streaks may be seen surrounding a protrusion in Figure 11c at a microscopic level. The development mode of both main and secondary dendrites causes the creation of these protrusions. The main dendrites develop parallel to the steel substrate, whereas the secondary dendrites align with the direction of heat flow. Solidification happens more frequently between secondary dendrites, producing glossy surfaces. Dendrites develop slowly and solidify last when they grow in the opposite direction due to heat flow suppression. If there is no more molten liquid available to compensate for solidification constriction, dull dendrites emerge, which may be seen microscopically as streaks [16,17]. Surface morphology (shiny, feathery, and dull spangle) influences surface micro-roughness, surface texture, interdendritic segregation, intermetallic compound precipitation, and, ultimately, corrosion resistance [16,17,27].

The morphology of glossy, feathery, and dull spangles differs in surface micro-roughness, surface texture, surface segregation, and precipitation, impacting corrosion resistance [16].

In the case of a thermally galvanized sample (HDG5), microscopic analysis using SEM revealed the presence of filamentous clusters of crystals on the surface of the deposited layer (Figure 12). These clusters are referred to as "outbursts" or "explosions" and represent localized growths of the intermetallic compound ξ. They occur because of a reaction that initiates after the piece is removed from the bath, as the growth is limited by the surface of the η layer. The ξ phases are narrow and elongated, with many of them being branched [15].

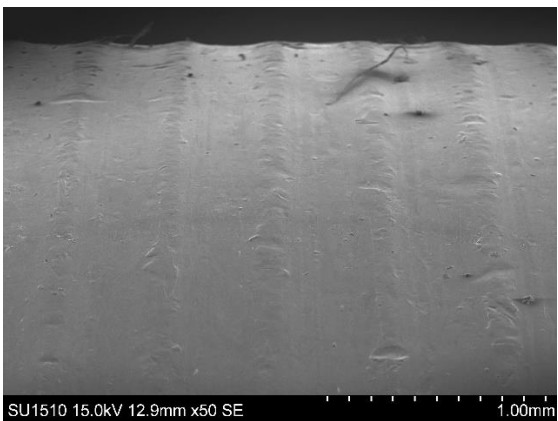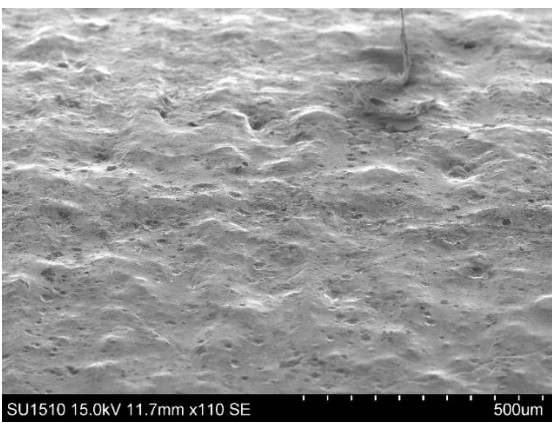

**Figure 12.** Image with filamentous clusters on the surface of the HDG5 sample.

Like the surface defect analysis performed on a galvanized sample, the surface analysis was conducted to quantify defects on samples with low roughness (HDG5) and high roughness (HDG1) of the steel substrate. The analyzed surface area was 1100 μm × 770 μm (847.000 μm$^2$ = 0.847 mm$^2$).

The surface defects (Figure 13), counted in 70 samples, have smaller dimensions, and mainly consist of cavities and protrusions. They have an approximately equiaxial shape. The total surface area of these defects was 1535.746 μm$^2$, which represents 0.18% of the total analyzed surface area of 847.000 μm$^2$ (Figure 13). The distribution of defects based on their maximum and average diameter in relation to the surface area, as well as the correlation between these sizes, is presented in Figure 13c. Most defects (objects) have a maximum diameter of less than 2 μm and a surface area of less than 2 μm$^2$.



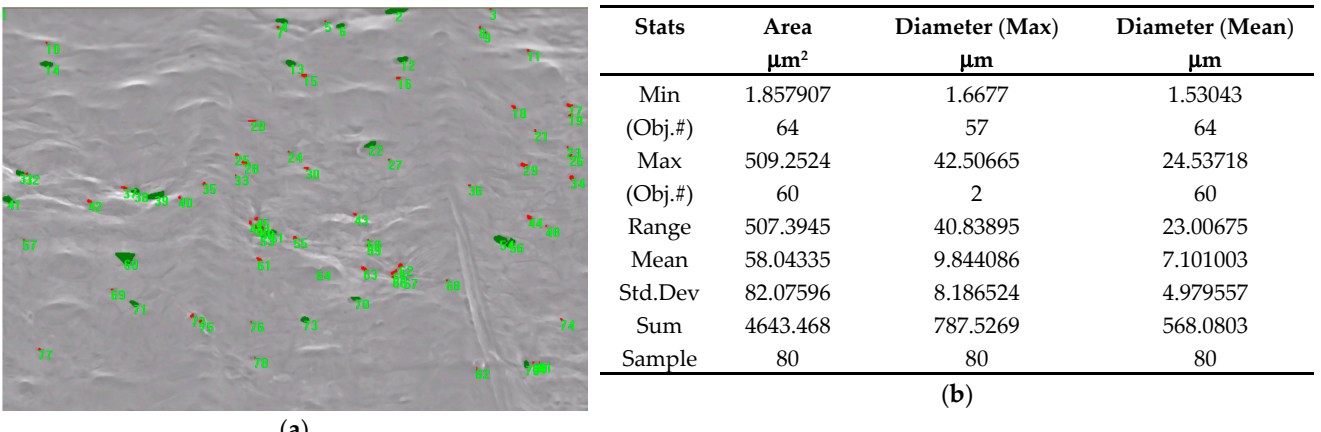

| Stats | Area<br>μm² | Diameter (Max)<br>μm | Diameter (Mean)<br>μm |
|---|---|---|---|
| Min | 1.857907 | 1.728704 | 1.544943 |
| (Obj.#) | 58 | 58 | 58 |
| Max | 116.6766 | 21.19095 | 13.71905 |
| (Obj.#) | 93 | 93 | 13 |
| Range | 114.8187 | 19.46224 | 12.17411 |
| Mean | 21.93923 | 6.275713 | 4.765422 |
| Std.Dev | 23.12095 | 4.288438 | 2.665828 |
| Sum | 1535.746 | 439.2999 | 333.5796 |
| Sample | 70 | 70 | 70 |

(b)

(c)

**Figure 13.** Surface analysis of HDG5 for quantifying surface defects: (**a**) numbering of defects; (**b**) measured values; (**c**) histograms of the defect diameter to defect surface area ratio.

For the HDG1 sample (high substrate roughness), the surface defects (Figure 14) observed on the analyzed surface (80 objects—Figure 14b) are small and mainly consist of cavities with a surface area below 1.86 μm². The total surface area of these defects was 4643.468 μm², representing 0.55% of the total analyzed surface area of 847.000 μm² (Figure 14b). Most defects (objects) have a maximum diameter below 1.67 μm.

| Stats | Area<br>μm² | Diameter (Max)<br>μm | Diameter (Mean)<br>μm |
|---|---|---|---|
| Min | 1.857907 | 1.6677 | 1.53043 |
| (Obj.#) | 64 | 57 | 64 |
| Max | 509.2524 | 42.50665 | 24.53718 |
| (Obj.#) | 60 | 2 | 60 |
| Range | 507.3945 | 40.83895 | 23.00675 |
| Mean | 58.04335 | 9.844086 | 7.101003 |
| Std.Dev | 82.07596 | 8.186524 | 4.979557 |
| Sum | 4643.468 | 787.5269 | 568.0803 |
| Sample | 80 | 80 | 80 |

(b)

(a)

**Figure 14.** *Cont.*

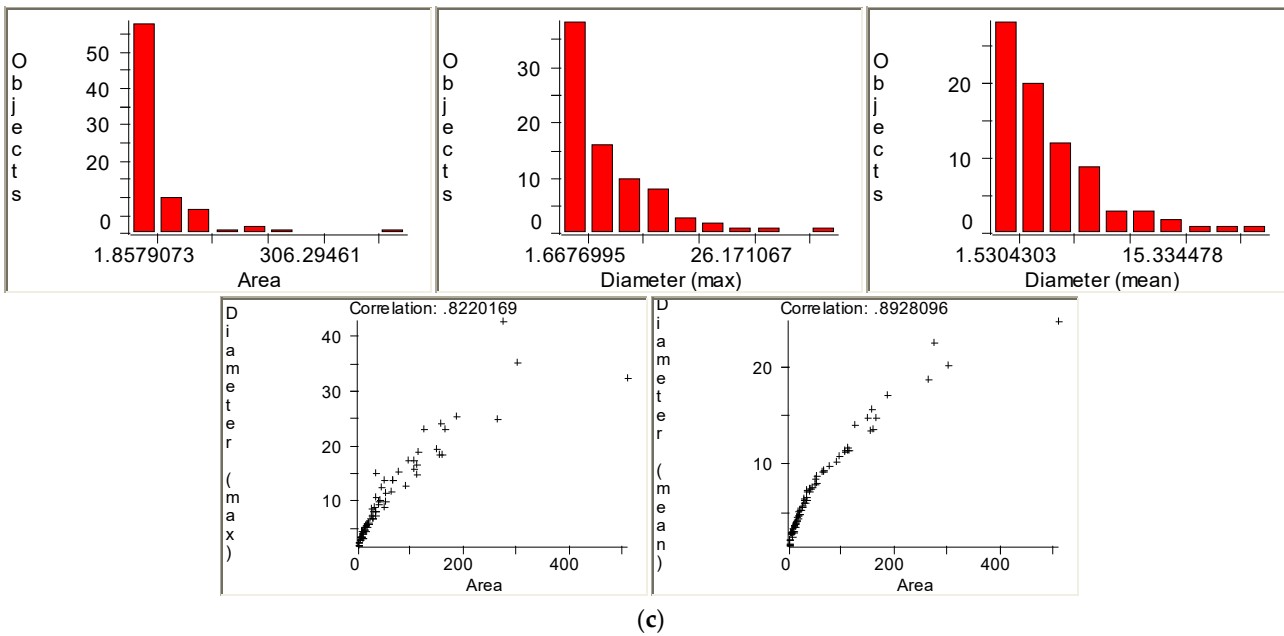

(**c**)

**Figure 14.** Surface analysis of HDG1 for quantifying surface defects: (**a**) defect numbering; (**b**) measured values; (**c**) histogram of the defect diameter/surface area ratio.

Surface analysis was carried out using ImagePro Plus software to examine the profile and topography in each direction. Figures 15 and 16 depict the results.

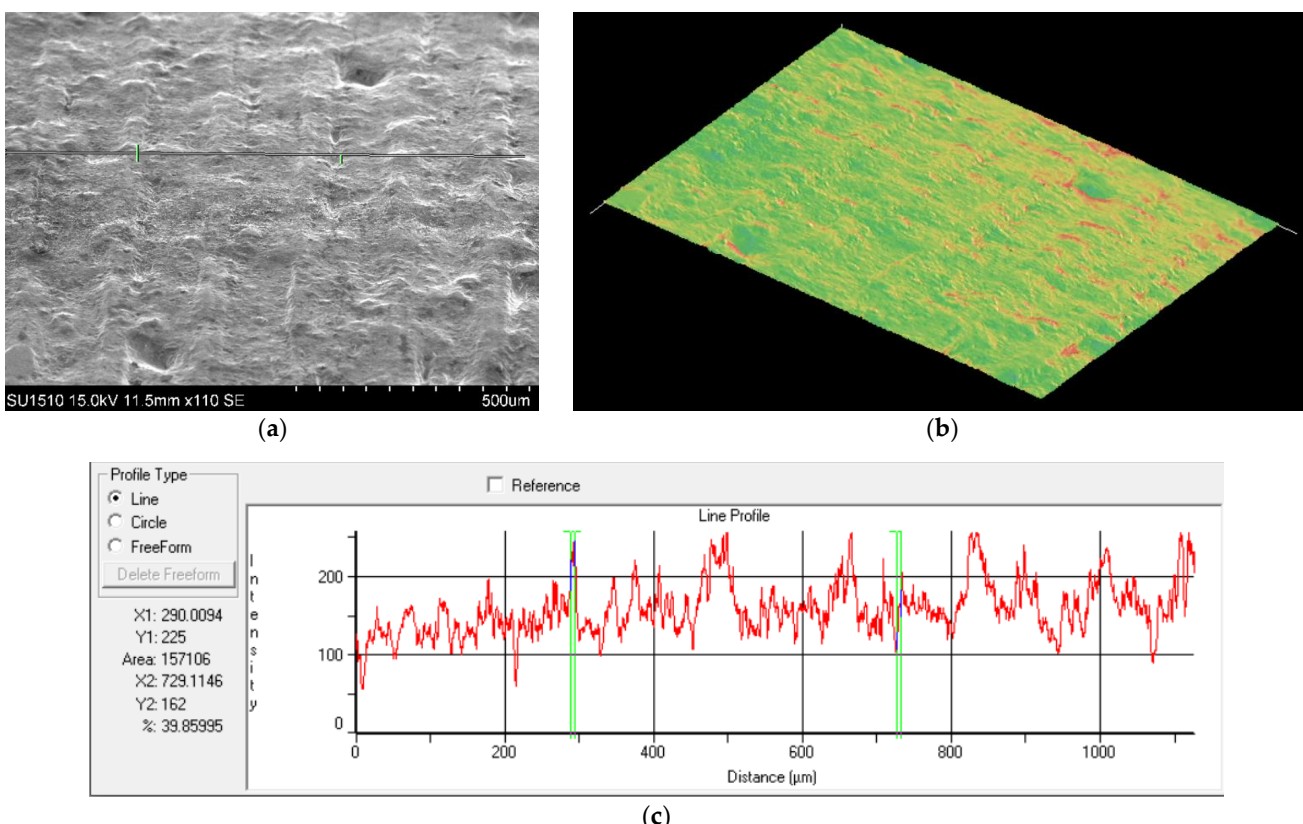

**Figure 15.** Topography and line profile of HDG5 sample (**a**) SEM image; (**b**) topography; (**c**) line profile.

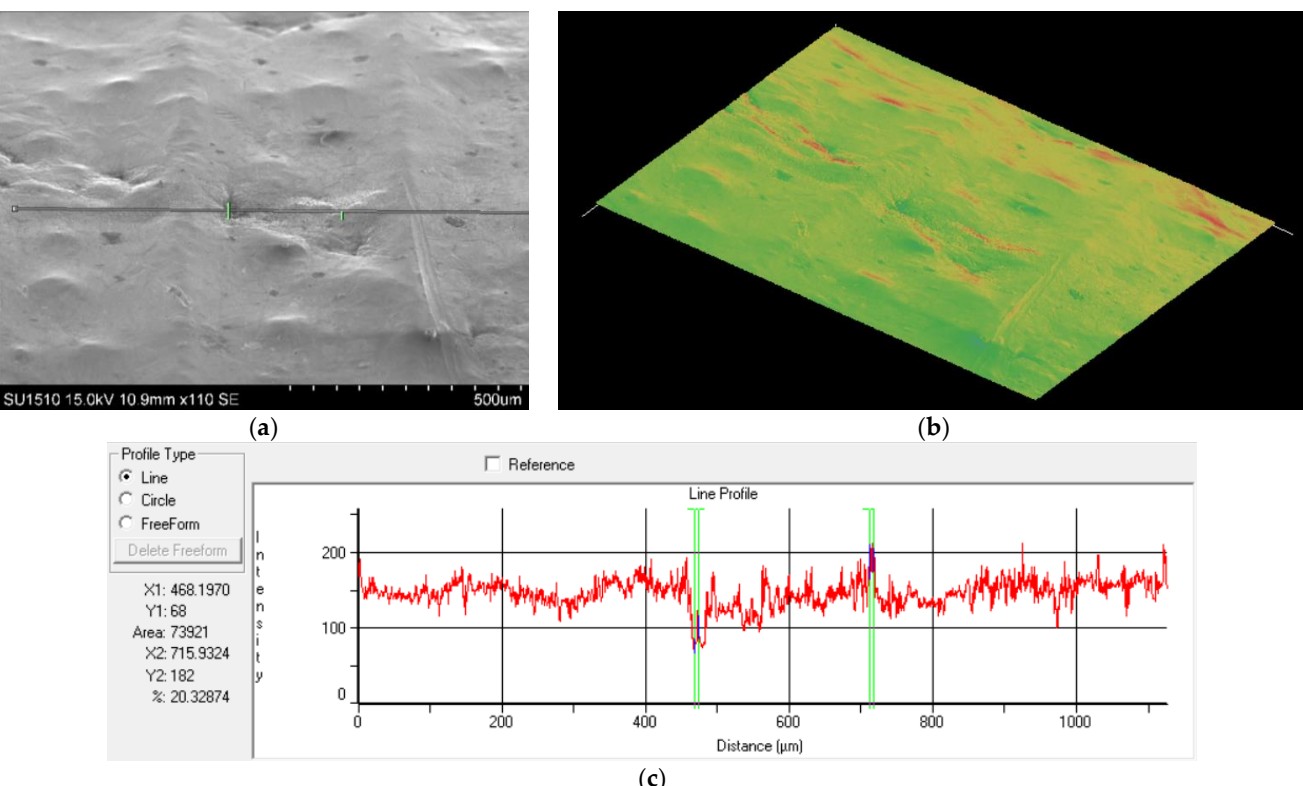

**Figure 16.** Topography and line profile of HDG1 sample (**a**) SEM image; (**b**) topography; (**c**) line profile.

By comparing the topography and linear profile of the HDG1 and HDG5 samples (Figures 15 and 16), it can be observed that the HDG5 sample with lower steel substrate roughness has a much higher roughness compared to the sample with higher substrate roughness (Figures 15 and 16) and significantly higher roughness compared to the electrolytically coated samples (GE1 and GE5).

For the cross-sectional analysis of the coating layer, the specimens were longitudinally sectioned. The sectioning process involved grinding with abrasive paper of various grit sizes and polishing on a cloth with alumina powder. The interaction zone between the deposited layer and the substrate in the cross-section of the galvanized samples was investigated using both optical and electron microscopy.

The GE, HDG, and HDGC samples were examined under an optical microscope using white light (LED) for macroscopic and microscopic analysis. At low magnifications, Figure 17 shows cross-sectional images of the finely machined specimens—GE1 sample (with low steel substrate roughness—Figure 17a) and the maximum substrate roughness GE5 sample (from the series of 5 specimens—Figure 17b), where the profile and configuration of the coating layer following the substrate profile can be observed.

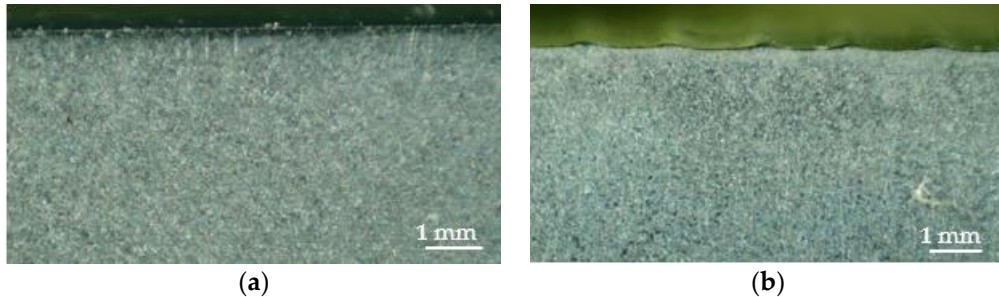

**Figure 17.** Cross-sectional images of GE specimens on a fine substrate (1–4.5×) (**a**) and on a coarse substrate (5–4.5×) (**b**).

Cross-sectional images of specimens galvanized by immersion and centrifugation are shown in Figures 18 and 19. Figure 18a depicts the investigation of the deposited layer on the cross-sectional area of the specimen with a very low steel substrate roughness (achieved by cutting the surface).

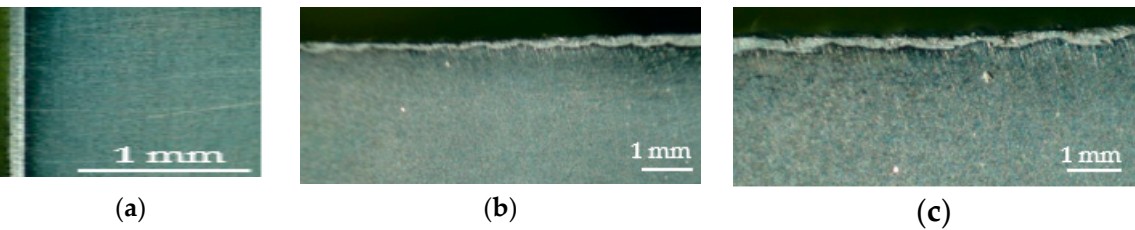

**Figure 18.** HDG section image (**a**) very fine-end (5–4.5×), (**b**) s. fine (5–4.5×), (**c**) coarse (1–4.5×).

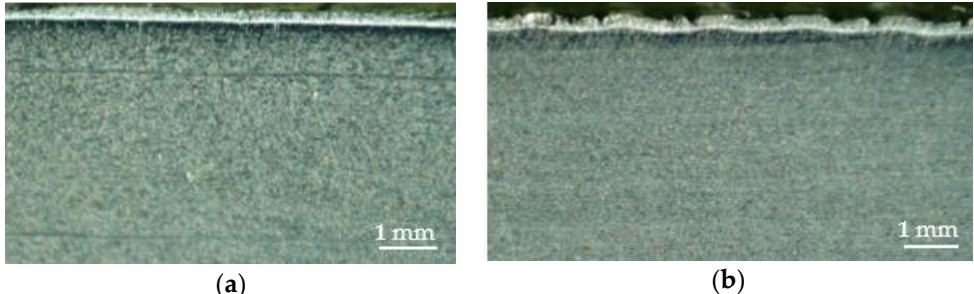

**Figure 19.** Image of HDGC specimen section: (**a**) fine turned (HDGC1), (**b**) rough turned (HDGC5).

For the immersion and centrifugal galvanized specimens (Figure 19), the surface roughness is much higher compared to the immersed specimens, as observed through roughness measurements, and confirmed by the cross-sectional analysis of the coating. While in the other galvanizing methods (GE and HDG), the cross-section of the coating showed a relatively continuous surface profile at low magnifications, roughly following the profile of the steel substrate, in the case of the HDGC specimens, there are significant variations on the surface of the coating with increasing substrate roughness, as highlighted in Figure 19b.

The sectioned specimens were further analyzed through optical and electron microscopy to observe the microstructure at the coating interface.

In the case of section analysis at high magnification (optical and SEM), the coating layer is distinctly observed, and the profile of the steel substrate interface at the Fe/Zn interface is approximately parallel to the outer layer profile (Figure 20). The plating bath was composed of high-purity zinc (99.97%), without significant concentrations of Al (below 0.1%) that would influence the formation of Fe-Zn intermetallic phases.

The typical morphology of a galvanized steel coating can be seen in Figure 21, specifically in the HDG1 specimen (with fine roughness), where the identified phases (gamma, delta, zeta, and eta) are present. The η phase corresponds to nearly pure Zn (usually 0.03% to 0.08% Fe [28]), and it is situated above the intermetallic compounds ζ—FeZn13 (94% Zn) and delta-FeZn7 (88% to 93% Zn). These intermetallic phases ζ and delta have a columnar structure and are brittle. The intimate atomic-level interface with the steel is represented by a very thin brittle phase, gamma (FeZn3) [28]. The microstructure of the HDG5 specimen (Figure 22) with high substrate roughness is similar in terms of phases to the microstructure of the HDG1 specimen, with a higher proportion of phase crystals in the section.

Impurities present in the surface layer can influence the coating. In the case of steel plates in a recoated condition, nanostructured oxides can develop on the surface due to the interaction of oxygen with elements such as Si or Mn which are present in the steel as impurities (at various concentrations) [29]. These oxides have poor wettability, leading to

reduced wettability of the steel substrate in the respective area with liquid zinc. The wetting contact angle varies between 80–130°, depending on the number of impurities [29]. The specimens were subjected to a turning operation after the annealing process, so the layer of nanostructured oxides was removed. However, the presence of oxides in the bulk material, resulting from machining operations, can reach the surface of the specimen (which became the substrate for galvanizing) through transcrystalline sectioning or chip detachment at the grain boundary. Surface roughness can affect the apparent contact angle. The size and distribution of surface oxides influence the wetting properties of steels [29].

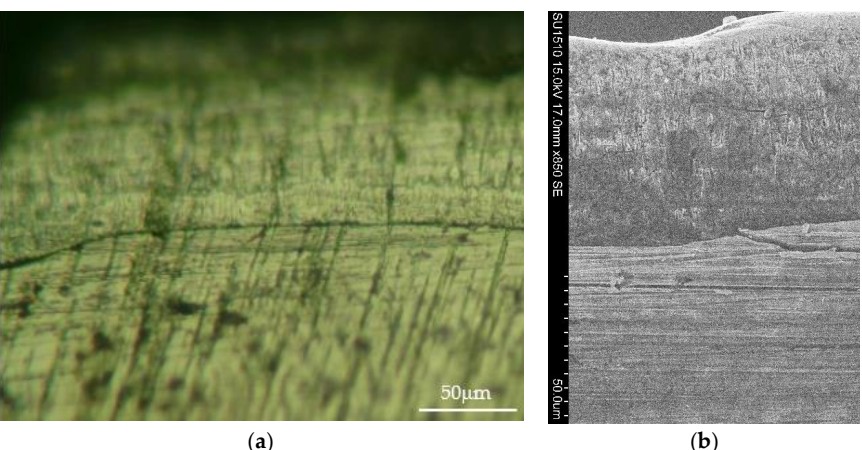

(**a**)　　　　　　　　　　　　　　　(**b**)

**Figure 20.** Section image of HDG5 specimen, polished (**a**) optical; (**b**) SEM.

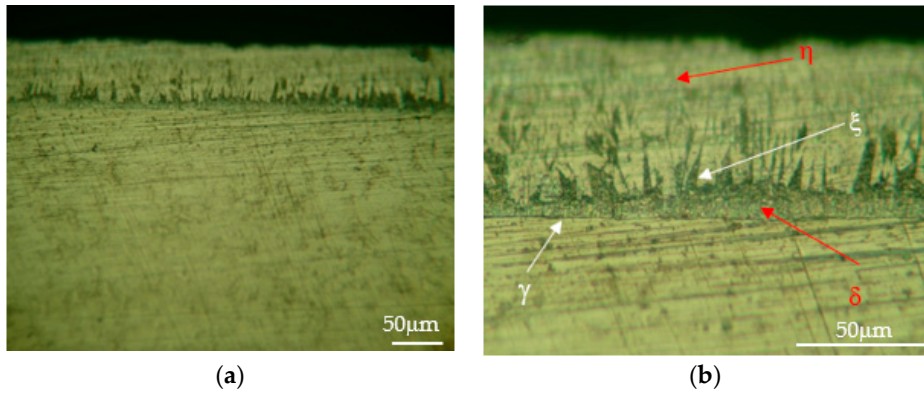

(**a**)　　　　　　　　　　　　　　　(**b**)

**Figure 21.** Section image of HDG1 specimen, etched with nital (**a**). The microstructure in section of sample HDGS1: (**a**) microstructure with pearlitic ferrite substrate, (**b**) microstructure with the intermetallic compounds in the coating layer.

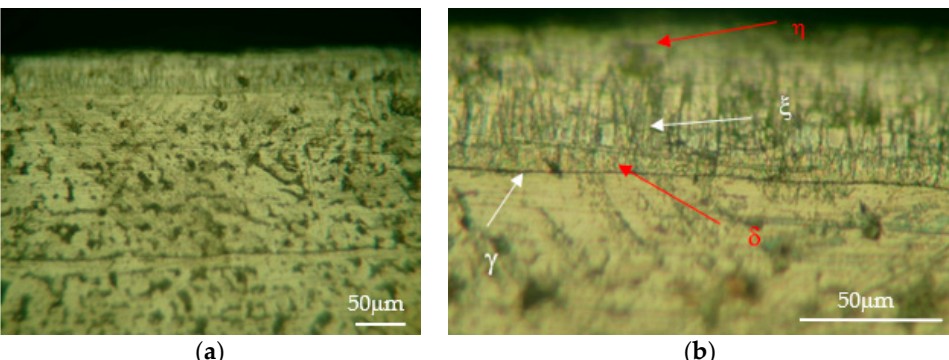

(**a**)　　　　　　　　　　　　　　　(**b**)

**Figure 22.** HDGC1s. The microstructure in section of sample HDGS1: (**a**) microstructure with pearlitic ferrite substrate, (**b**) microstructure with the intermetallic compounds in the coating layer, nital attack.

The reaction between Zn and Fe forms a series of intermetallic layers of Fe-Zn alloy, metallurgically bonded to the base metal, with properties superior to those of iron. A relatively high proportion of the zeta phase in the iron–zinc alloy can lead to localized microcracking, as observed in Figure 23. The occurrence of microcracks has been highlighted between the gamma and delta phases, with the delta phase being the intermetallic compound with the highest hardness among the Fe-Zn compounds. Under the action of strong impact, torsion, or excessive bending, these microcracks can lead to the delamination of the zinc coating.

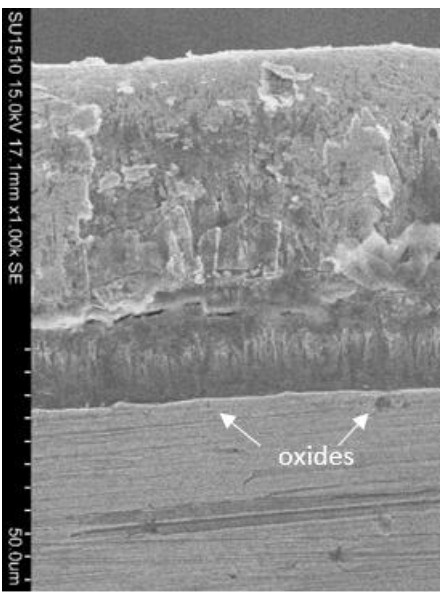

**Figure 23.** Crack in zinc layer between delta and zeta phases in thermally immersed sample HDG5.

Microcracks have been identified at the intermediate phases formed between Fe and Zn (Figure 23). These primary cracks are mainly caused by the growth of the film, primarily due to differences in the coefficients of linear thermal expansion and elasticity between the steel and zinc [28]. The gamma and delta layers, being brittle layers, are prone to the formation of primary microcracks. The composition of the metal bath influences the formation of brittle layers, and as the thickness decreases, the length of the primary cracks is reduced. The quantity or number of cracks increases as stresses are applied to the specimens. If the applied stress is low, the cracks do not reach the steel surface, and the coating maintains its protective function. However, for high and cyclic stresses, the cracks can coalesce and propagate to the surface, leading to the deterioration of the coating. The effect of residual compressive stress on the substrate surface acts as an inhibiting layer for the growth of the coating and prevents atomic diffusion between Fe/Zn interfaces [30,31].

Another encountered defect is the presence of microcraters on the surface (Figure 24). In this area, a decrease in the thickness of the gamma phase layer is observed from the crater's edge towards the center, accompanied by the absence of other intermetallic phases and the zinc layer.

When applied to zinc-coated specimens, the number of cracks grows in proportion to the applied stress. When these stresses are modest, the fractures do not reach the substrate's surface, and the zinc coating performs its corrosion-protection role. However, at high and cyclic pressures, the initial fractures (microcracks) might connect and spread to the surface [28]. Immersion-galvanized components subjected to tensile or bending stresses may fracture due to the presence of big grains in the coating, which has a characteristic casting structure with dendritic morphology and significantly larger grains than those in cold-processed metal. Visible fractures appear often, and the coating is susceptible to intergranular cracking or trans-granular cleavage cracking. Because of zinc's hexagonal crystalline structure and coarse grain structure, the coating is prone to both intergranular

and trans-granular cracking during bending or forming processes. Because of the cathodic protection given by zinc, crack development has a limited influence on the coating's lifetime and corrosion resistance. Corrosion of the steel substrate does not occur until the zinc layer in any specific portion of the surface is entirely eaten [28].

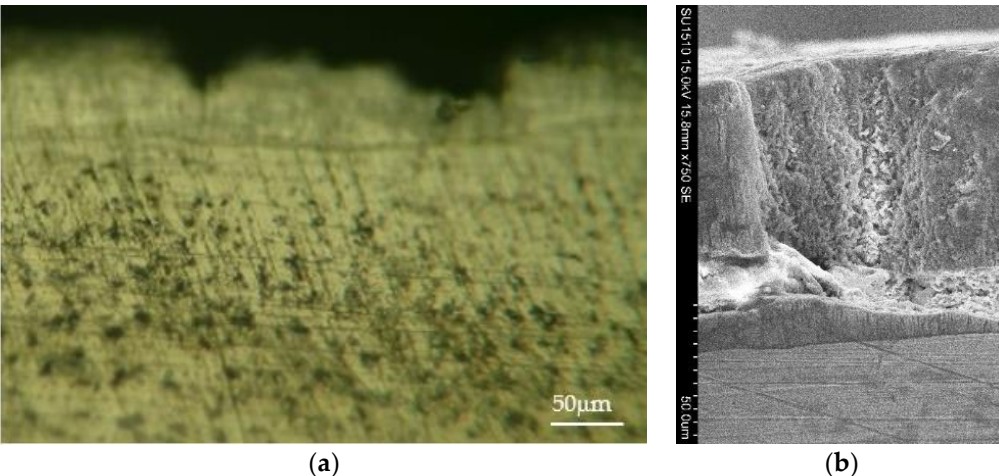

(**a**)                                                           (**b**)

**Figure 24.** Sectional image of sample HDG5 around a crater (cavity); (**a**) optical image; (**b**) SEM image.

The existence of oxide patches (films) on the steel substrate, which delays the development of the coating, might explain the creation of cavities (craters). The zinc coating near the crater's base was incredibly thin. This was seen in the instance of a steel coating with 1.7% Si and 2.7% Mn, where oxygen was detected, indicating the existence of silicon and manganese oxides at the crater's base. These oxides hampered the diffusion of Zn and Fe, resulting in the development of craters [32]. Furthermore, the presence of oxides (small oxide grains among the crystalline grains) within the surface layer of the steel substrate was found (Figure 25), which is responsible for the creation of craters on the coated surface [32]. The appearance of striations (grooves) on the coating surface, which rise above the submicron oxide layers situated at the grain boundaries of the steel substrate, can be explained in the same way. The craters' occurrence can be explained by inadequate wetting of the surface by certain kinds of exogenous oxides, resulting in "empty spots" in the zinc layer [21]. The dissolution of the oxide film and slower development of the zinc layer at the base of the groove, as well as the existence of coalesced oxide crystals at the base, explain the decreased thickness of the coating at the base of the groove [21].

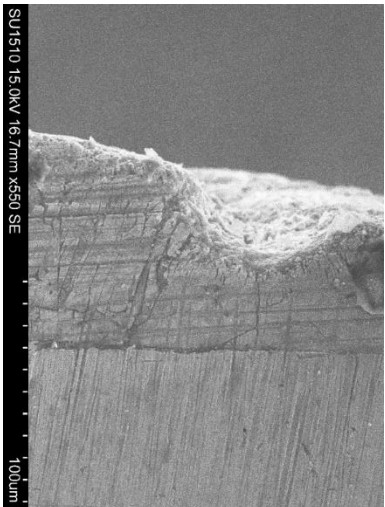

**Figure 25.** Small cavity (indentation) in the zinc layer in thermally immersed centrifuged sample HDGC5.

Microcavities (Figure 25) can appear on the surface of the η phase, but they do not reach the ξ phase.

The adherence of a zinc coating on DP steels can be enhanced by preventing the formation of Fe-Zn complexes and oxides at the zinc coating–steel substrate interface. A sufficient Al concentration in the zinc bath, as well as a decrease in oxide formation during annealing prior to zinc coating deposition, are advantageous [21].

Impurities and oxide compounds present in the surface layer tend to segregate at the η dendrite boundaries, contributing to increased roughness and achieving a homogeneous surface to improve the corrosion resistance of the coating [32].

## 5. Findings and Practical Implications

In atmospheric circumstances, the compactness and toughness of corrosion products generated on the surface of zinc give strong corrosion resistance to zinc coatings. Corrosion products that are first generated grow more sticky and denser with time. Only in the crevices where the zinc surface is not sealed by corrosion agents can further corrosion occur.

In most environments, the anodic characteristic of zinc provides cathodic protection to steel. Minor discontinuities or tiny exposed steel portions caused by holes or cut edges are shielded from corrosion by zinc's sacrificial cathodic protection. The corrosion products produced by this operation provide further protection [33].

The galvanized coating protects steel against corrosion (rusting) by acting as an environmental barrier and sacrificing itself to offer cathodic protection. The galvanized layer provides barrier protection, which is strengthened by the formation of a thin, strongly adherent layer of zinc corrosion products on the coating surface. Upon initial degradation of a freshly galvanized surface, $ZnO$ is formed and transformed into $Zn(OH)_2$ in the presence of moisture. Subsequent reactions with $CO_2$ in the air result in the formation of basic $ZnCO_3$, which is relatively insoluble and inhibits further corrosion. The gray patina typically associated with deteriorated galvanized coatings is the result of this thin layer of basic $ZnCO_3$. In marine environments, due to salt spray, zinc hydroxy chloride $(Zn_5(OH)_8C_{12}.H_2O)$ can also form, which represents a significant compound on the surface of the samples [28,34–36].

The assessment of atmospheric corrosion on electro-galvanized (EG), hot-dip galvanized (HDG), and hot-dip galvanized alloy (Galvanneal-GA) steels was conducted using accelerated field and cyclic tests, as presented in the work [37]. Three years of outdoor corrosion testing, accelerated by salt spray and SAE J2334 and GM 9540P cyclic exposure, recorded a sudden increase in the average corrosion advancement and maximum corrosion penetration of GA steel with a $40\,\mathrm{g/m^2}$ zinc coating, resulting in inferior corrosion resistance compared to EG and HDG steel, reaching complete thickness loss after three years.

In the automotive industry, there is a tendency to apply various post-galvanizing treatments to improve corrosion resistance, such as painting. A zinc coating thickness of approximately 8 μm over which a paint layer was applied exhibited corrosion resistance 16 times higher than that of cold-rolled, annealed, and painted sheets in the salt spray test [38].

Through corrosion investigations, including salt spray, humidity chamber, air exposure, and electrochemical testing, detailed research has been conducted to determine the major cause of the mottling (darkening) problem of hot-dip galvanized steel sheets [39]. Shiny regions were more resistant to corrosion than matte (dull spangle) parts. Higher concentrations of some elements (Pb, Sb, etc.) create a galvanic cell with zinc, causing galvanized sheets to darken prematurely. The passivation solution was composed of chromic acid and sulfuric acid, where chromic acid is a strong oxidizing agent that passivates a metal surface without forming a chromate film, and it worked very well in reducing the problem of early darkening of galvanized sheets by selectively dissolving lead and antimony [39].

Because of the absence of accessible and precise inspection methods and the widely variable morphological features seen in Figures 26–28, the morphology of corrosion prod-

ucts is more difficult to identify than their chemical properties. Corrosion products are a plethora of zinc-based compounds generated in many types of atmospheres, but only specific compounds dominate in each atmosphere. Oxides and chlorides are the most common corrosion products in the salt spray corrosion testing samples (Figure 26c).

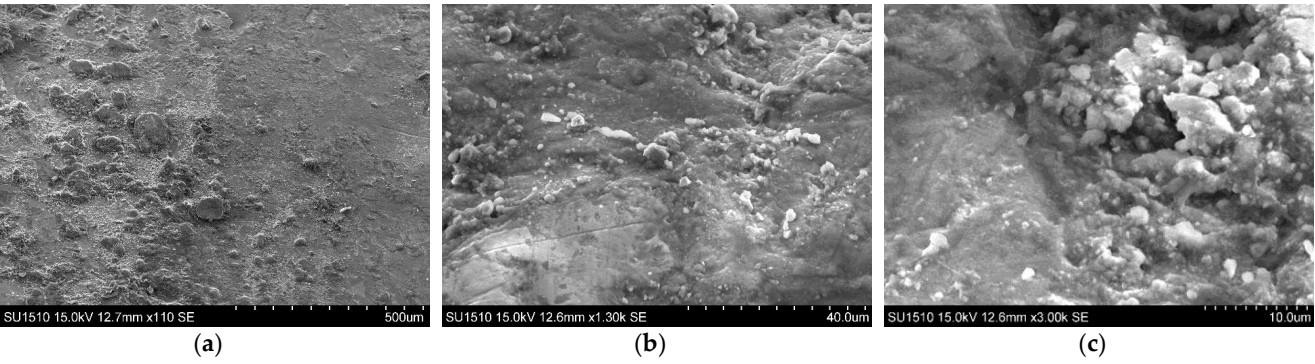

**Figure 26.** GE (1) with low roughness corroded (**a**–**c**).

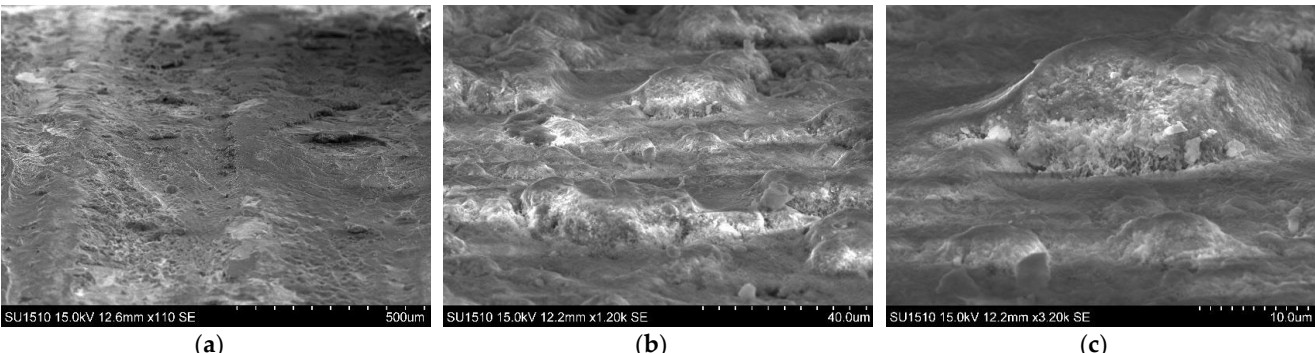

**Figure 27.** GE (5) with high roughness corroded (**a**–**c**).

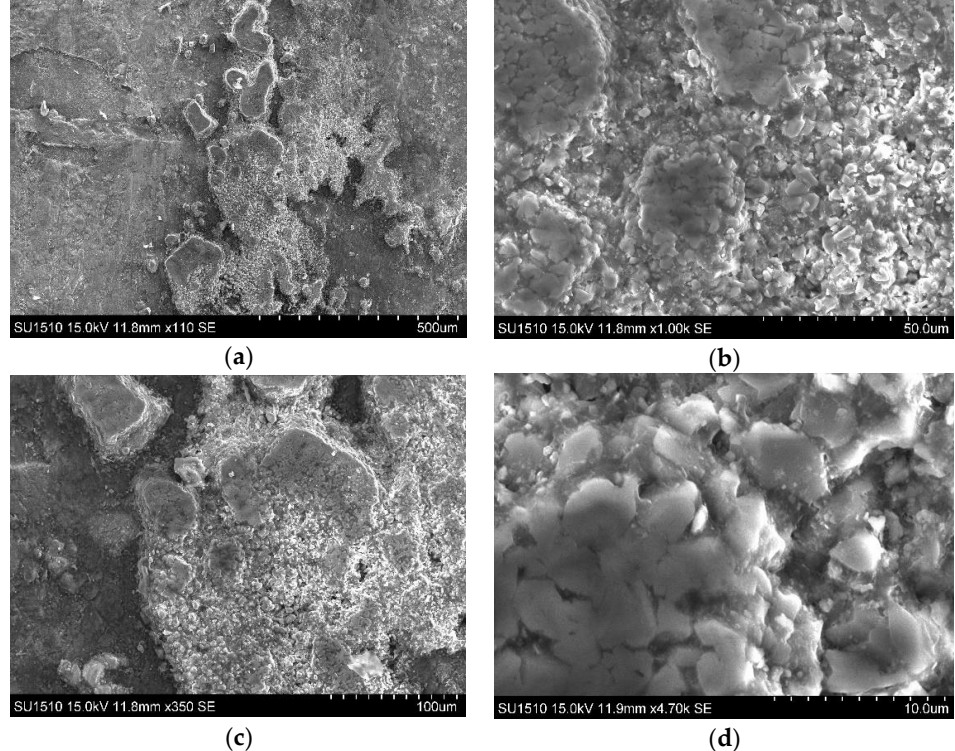

**Figure 28.** HDG (1) with low roughness corroded (**a**–**d**).

Deep ellipsoid, shallow ellipsoid, groove, and saddle forms are the four types of corrosion pits on the surface of steel wires [30].

Figure 26a shows globular-shaped crystals with rounded edges on the surface of electrolytically produced zinc (GE1 sample) subjected to corrosion, and Figure 26b shows corrosion products on the surface of and between these zinc crystals. The discrepancy in roughness value might be related to changes in the coating's microstructure [40]. When GE5 samples (with increased roughness of the steel substrate) are electrogalvanized (Figure 24), there is less corrosion product on the surface than with the GE1 sample. Figure 27c crystals had an acicular shape, confirming that after the zinc layer dissolving, there was a cluster of filiform phases identical (but much bigger) to what was seen in Figure 26.

The current density during electrolytic deposition has an influence on the corrosion resistance of the coating. Deposition at 150 mA/cm$^2$ results in better corrosion resistance in the pyramidal plane (101) and (103) compared to densities of 50 and 250 mA/cm$^2$, respectively. The film-substrate constant-phase coefficient is high, which resists the dissolution of the film in the solution and leads to better corrosion resistance.

The outermost and innermost layers both played important roles in galvanic activity [40]. The measured potential changes during anodic processing show that the zeta and phases have greater corrosion resistance than the top zinc layer [40]. The inner layers of Fe-Zn intermetallic compounds outperformed the outer layers in terms of galvanic performance. The total zinc concentration of the coating was not an absolute criterion of the coating's protective potential. The corrosion resistance of the coating is determined not only by the amount of zinc in the outer layer, but also by the amount of zinc in the intermetallic phases [40].

In the case of a hot-dip galvanized sample (HDG5), microscopic SEM analysis revealed clusters of filiform crystal outbursts on the surface of the deposited layer (Figure 26). These clusters, known as "outbursts", are narrow and elongated filiform phases, more corrosion-resistant than the η phase or filiform compounds resulting from the corrosion of the ζ phase [15].

Various cavities with rounded edges, corroded grains with rounded edges, and different filiform corrosion products can be seen on the surfaces of immersion zinc-coated samples (HDG) and immersion and centrifuged zinc-coated samples (HDGC) exposed to corrosion testing (Figure 27b,c). Because the initial roughness of these samples was substantially higher than that of electrogalvanized samples, the amount of surface corrosion varied greatly. Variations in the microstructure inside the coating, the existence of various intermetallic phases in the coating, and a different fluctuation in the thickness of the Zn layer (eta) [40] might all be linked to the roughness value [40].

The HDG1 sample with low substrate roughness exhibited more intense corrosion and developed a series of reaction products on its surface (Figure 28). This can be explained by the initial roughness of both the coating layer and the steel substrate, as well as the growth of smaller-sized dendritic crystals of η phase. Underneath this layer, columnar crystals of ξ phase formed, acting as the cathodic region, and leading to faster anodic dissolution of the η phase. This resulted in the formation of corrosion products in various forms, with a higher proportion being represented by prismatic crystals (Figure 28).

The η phase in the corrosion process, due to the difference in corrosion potential, acts as an anode at the interface between the η and ζ phases, while the ζ—FeZn13 phase acts as a cathode, forming a galvanic corrosion couple [41–43] (Figure 29). During corrosion, the Zn phase preferentially corrodes, while the ζ—FeZn13 phase, the cathode, remains unchanged. With the increasing corrosion cycle, the η-Zn phase is corroded and detached, forming typical galvanic corrosion [28]. In Figure 27, a cavity with a diameter of approximately 50 μm can be observed, where the η phase layer has been corroded down to the Fe-Zn intermetallic phases. Once the η layer is dissolved at the base of the crater, the ζ phase crystals become visible, which have higher corrosion resistance than the η phase (Figure 30c). On the surface of the sample and at the edge of the cavity, accumulations of corrosion

products can be observed. It can be stated that Figure 30 exhibits a "difference in level" in a corroded area.

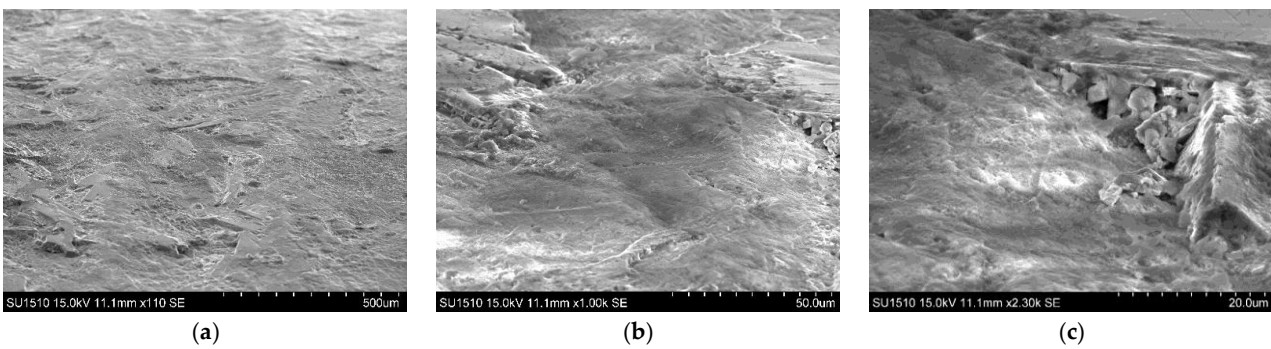

**Figure 29.** HDG (5) with high roughness corroded (**a**–**c**).

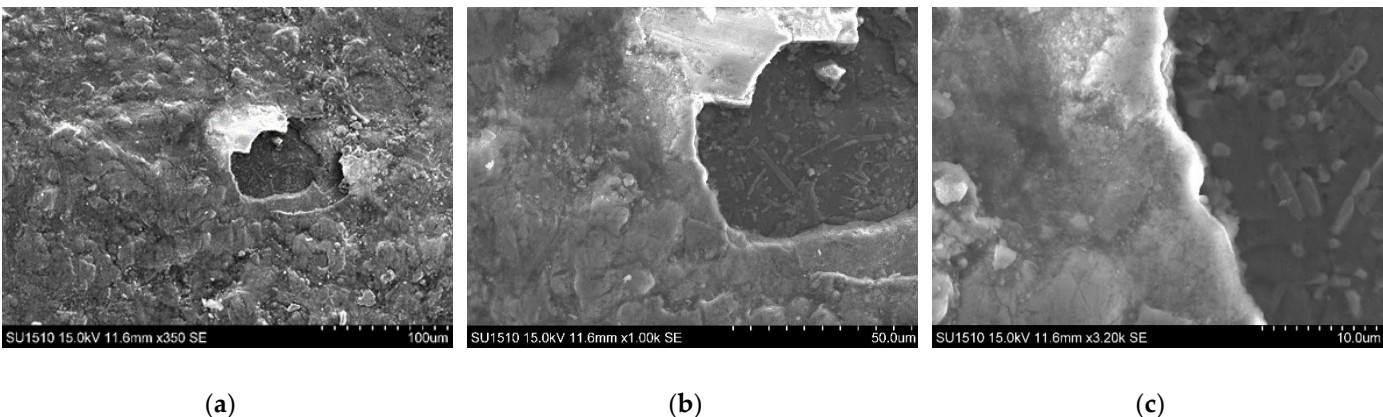

**Figure 30.** SEM image in sample HDG5—cavity ("crater") in corroded sample at various magnifications (**a**–**c**).

Figures 31 and 32 illustrate the surface examination of the submerged samples (HDG1, HDG5) using the ImagePro Plus program to highlight the corrosion mode (corroded regions) and corrosion compounds (areas with distinct types of corrosion compounds).

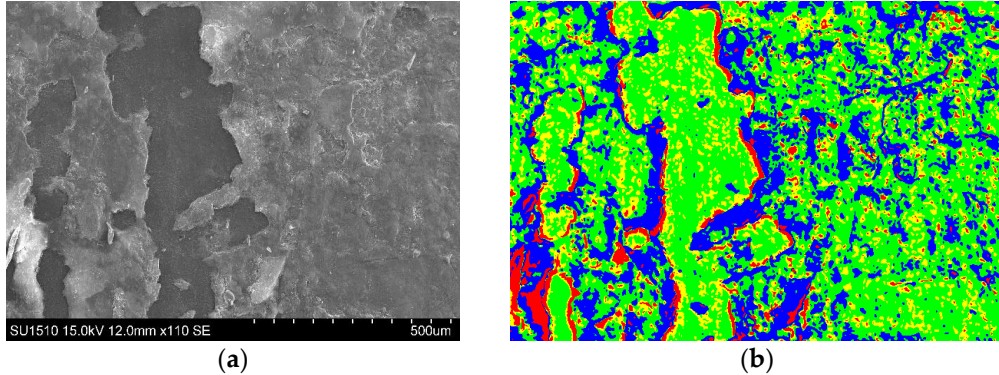

**Figure 31.** Corrosion surfaces sample HDG1. (**a**) SEM surface image; (**b**) map of corroded portions.

The analyzed surface area for sample HDG5 was 847.000 $\mu m^2$. Different colors were assigned to corroded areas (green) and areas with corrosion compounds, with red representing concreted compounds on the surface, yellow representing isolated crystals, and blue representing less attacked areas. The analysis results for the analyzed surface are presented in Table 4.

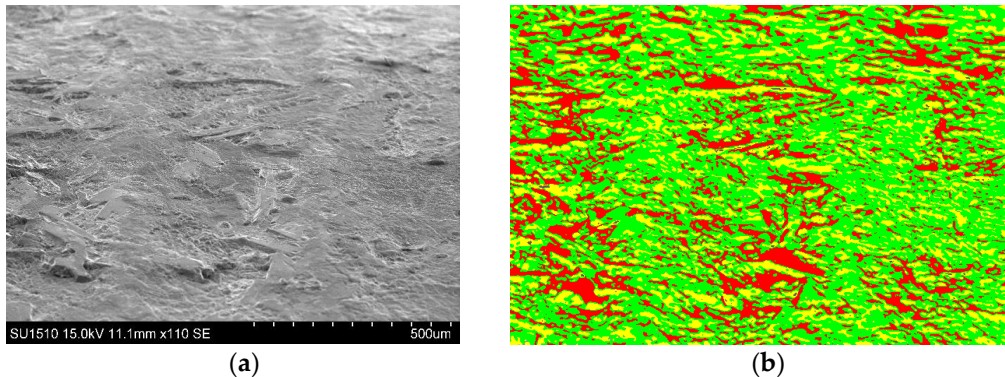

**Figure 32.** Corrosion surfaces sample HDG5. (**a**) SEM surface image; (**b**) map of corroded portions.

**Table 4.** Proportion of corroded areas in relation to area analyzed.

| Range | Objects | % Objects | Area % |
|---|---|---|---|
| 1 (red) | 924 | 26.6436 | 7.20622 |
| 2 (green) | 292 | 8.419839 | 56.19326 |
| 3 (blue) | 688 | 19.83852 | 24.6255 |
| 4 (yellow) | 1564 | 45.09804 | 11.97503 |

The measured values show that more than 56% of the surface in HDG5 is corroded.

In the case of the HDG1 sample, the deeply corroded areas were much fewer. Green was assigned for the corroded area, yellow for corrosion products, and red for the shallowly corroded area (Figure 31b). The results of the analysis are listed in Table 5.

**Table 5.** Proportion of corroded areas in relation to area analyzed.

| Range | Objects | % Objects | Area % |
|---|---|---|---|
| 1 (red) | 972 | 34.62772 | 28.18978 |
| 2 (green) | 1509 | 53.75846 | 24.07276 |
| 3 (blue) | 326 | 11.61382 | 47.73746 |

The surface of sample HDG1 is covered with corrosion products over 54% of the surface, the deep corroded area being much smaller than in sample HDG5 due to the smoother sample surface (Figure 32a).

Very few rust spots were observed on the surface of the galvanized samples. Some spots were observed in areas with high amounts of corrosion products (Figure 33a), or in areas where the protective coating showed cracks; thus, the corrosive environment led to the dissolution of the coating and the appearance of rust (Figure 33b). Rust compounds can be FeOOH on the outside of the substrate and internally $Fe_3O_4$ compound [44–47].

Rust on steel is porous and adheres poorly to the surface of the steel substrate. Rust can increase corrosion by acting as a reservoir for water, hence, extending wetting time, or as a catalyst for oxygen reduction, as demonstrated on the surface of a zinc-coated sample using both optical and electron microscopy (Figure 33b).

Corroded components have an impact on their strength. The fracture morphology of corroded steel [48–53] revealed that as the corrosion rate increased, so did the mechanical parameters of the samples (loss in yield strength, elongation, and strength).

Due to the production of corrosion products on the steel surface, which gradually slows down the corrosion rate, the corrosion rates of the steel samples are indirectly related to the immersion duration [54].

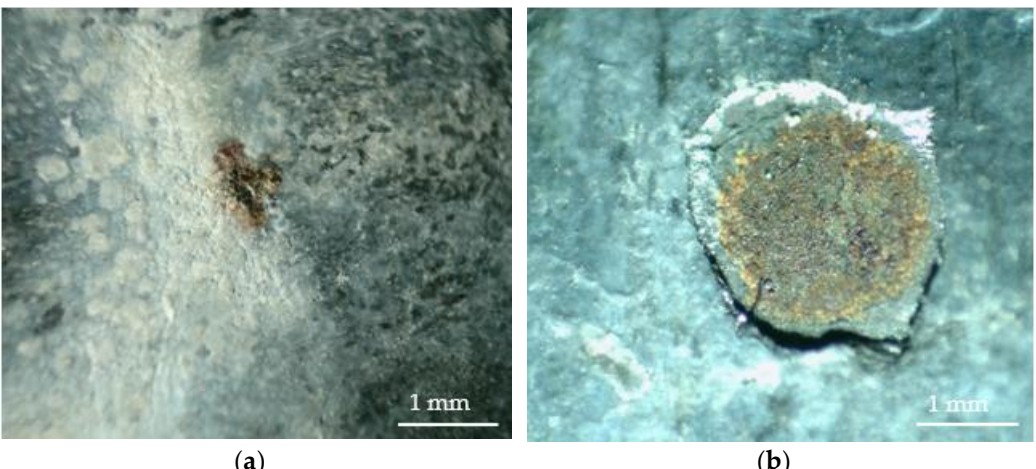

(**a**)  (**b**)

**Figure 33.** Corrosion cavity ("crater") in oxidized sample: (**a**) rust under the layer of corrosion compounds; (**b**) rust cavity in steel substrate.

## 6. Conclusions

This study's goal was to look at the corrosive behavior of zinc and steel coatings under various exposure scenarios. The mechanisms of corrosion and the generation of corrosion products were revealed by studying galvanized and corroded steel samples. Rust has minimal adherence to the steel substrate and can act as a catalyst for cathodic processes, extending the presence of moisture or accelerating oxygen reduction. As a result, the study met its aim of studying corrosion processes and identifying the components involved in the deterioration of zinc and steel coatings.

This study has greatly improved our understanding of corrosion processes and the interactions between zinc coatings and steel substrates. Through the study of galvanized and corroded steel samples, as well as an extensive literature investigation, the research shed light on the significance of the initial roughness of the coating and substrate in corrosion development. Furthermore, the production of zinc dendritic crystals inside the coating was discovered to influence the corrosion process. These results have increased our understanding of how zinc coatings interact with steel substrates, opening new avenues for the development of corrosion control techniques.

The research provided significant theoretical and practical advances. The mechanisms of corrosion have been identified, and the corrosion products produced on the surface of zinc and steel coatings have been characterized. These discoveries have contributed to the corpus of information on corrosion and assisted in the development of more realistic theoretical models. In practice, the research has yielded important information for the corrosion protection industry. The findings can be utilized to improve galvanization techniques as well as to develop more corrosion-resistant materials and technologies. As a result, this study has resulted in significant advancements in the field of corrosion prevention application.

Based on the data, many prospective study directions might be suggested. An important component is the investigation of the effect of various environmental conditions on the corrosion of zinc and steel coatings. This might require looking at how humidity, temperature, corrosive substance concentrations, and other environmental factors affect corrosion processes. It would also be interesting to investigate how the chemical composition of zinc coatings and the presence of alloying components affect galvanized coating corrosion resistance. These study directions might aid in the development of more efficient and long-lasting corrosion prevention technologies.

Finally, this research accomplished its original purpose of offering fresh knowledge and viewpoints on the corrosion of zinc and steel coatings. The acquired results have made major theoretical and practical contributions, which have proven useful in the corrosion

protection business. The suggested future research paths bring up new avenues for progress and innovation in the field of corrosion prevention.

**Author Contributions:** Conceptualization, G.I., S.R.-N., A.B.P. and A.M.T.; methodology, A.B.P. and A.M.T.; software, G.I. and S.R.-N.; validation, A.M.T. and A.B.P.; formal analysis, A.B.P.; investigation, G.I. and S.R.-N.; resources, A.B.P., S.R.-N. and G.I.; data curation, G.I. and S.R.-N.; writing—original draft preparation, A.B.P., G.I., A.M.T. and S.R.-N.; writing—review and editing, A.B.P., G.I., S.R.-N. and A.M.T.; visualization, S.R.-N., A.B.P. and A.M.T.; supervision, A.M.T., A.B.P., G.I. and S.R.-N.; project administration, A.B.P. and S.R.-N. All authors have read and agreed to the published version of the manuscript.

**Funding:** This research received no external funding.

**Institutional Review Board Statement:** Not applicable.

**Informed Consent Statement:** Not applicable.

**Data Availability Statement:** Not applicable.

**Conflicts of Interest:** The authors declare no conflict of interest.

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
