# Peer review of "Characterization and Corrosion Behavior of Zinc Coatings for Two Anti-Corrosive Protections: A Detailed Study"

_coatings, doi:10.3390/coatings13081460_

Round 1

Reviewer 1 Report

Alina et al. The purpose of this research is to examine the corrosion behavior of zinc coatings used for corrosion protection, specifically on S235 steel. The study highlights the importance of corrosion protection in industrial settings and the need for effective solutions. This work has significant implications for improving protective systems in industrial applications. I am glad to accept this work after major revisions.

1. Title can be shorten, personally suggestion, 'Characterization and Corrosion Behavior of Zinc Coatings for 2 Anti-Corrosive Protection: A Detailed Study'

2. Introduction: Please rephrase this part, especially for the examples of others' work. It can be emphasized and with powerful and insightful. 

3. All microstructure of samples should be with scales. Most ignored. Please check all and correct them. 

4. Fig. 4c, avoid the screenshots and make your data visual using data processing software, even MS EXCEL.

5. Fig. 7, the same with Q4. 

6. If possible, please summarize all SEM images that appeared in this manuscript, some can be transferred into the SI file. 

7. References: update the ref list with the latest version. 

Reviewer 2 Report

The review deals with the Characterization and Corrosion Behavior of Zinc Coatings for 

Anti-Corrosive Protection: A Detailed Study on Corrosion 

Product Formation, Performance Evaluation, and Influence of 

Deposition Parameters on S235 Steel. 

According to the reviewer, the paper is worth publishing at the Coatings Journal, 

but some corrections are needed and then the paper can be accepted for publication in the journal.

Additionally make the following corrections to the manuscript:

Comment 1

The authors should check the paper for spelling and typographical errors.

Line 98

Petit [3] performed

The authors should replace

Petit et al. [3] performed

Line 102

Bolzoni [4] emphasized

The authors should replace

Bolzoni et al. [4] emphasized

Line 106

Le and colleagues evaluated

The authors should replace

Le et al. evaluated

Line 110

Morcillo and Daz investigated

The authors should replace

Morcillo et al. investigated

Line 114

Li and Han focused

The authors should replace

Li et al. focused

Line 117

Galedari and colleagues

The authors should replace

Galedari et al.

Line 124

Pokorn investigated galvanized

The authors should replace

Pokorny et al. investigated galvanized

Line 124

Abd El-Lateef and colleagues investigated

The authors should replace

Abd El-Lateef et al. investigated

Line 132

Yan looked on the effect

The authors should replace

Yan et al. looked on the effect

Line 358

duration [21,22,23].

The authors should replace

duration [21 - 23].

Line 760

was less rough (Fig. 29a) 

The authors should replace

was less rough (Fig. 29a). 

Comment 2

Table 1

The authors should give more details for the Table 1 (authors experiments or sypplier's data).

Comment 3

The authors should explain why they used two times the "(22 mm)".

Comment 4

Table 2

The values from the Table 2 are the mean values of the roughness (deviation?)?

Comment 5

Line 222

A Hitachi SEM was used

The authors should give more details for the using equipment (type, model).

Comment 6

Line 233

The authors should insert a description for a) and b) 

and the authors should insert the magnification.

Comment 7

Line 252

d) roughly (5 - 4.5x, Fig. 23) stereoscopic images. 

The authors should check if the number of the Figure is right (23?).

Line 305

than 4m, and the computed

The authors should check if the unit is right (m? or μm?).

Line 306

10µm (Figure 3-1.c).

The authors should check if the number of the Figure is right (3-1.c?).

Line 560 

Zn (Figure 23a). These primary

The authors should check if the number of the Figure is right (23a? - there is no Figure 23a).

Line 572

Figure 23. Crack in zinc layer between delta

The authors show the oxides (and no cracks) in the Figure

Comment 8

Figure 11

If it possible, the Figure must be accompanied on the same page as the Figure's title.

Comment 9

Line 621

5. Findings and practical implications 

The authors should consider if they use the 

5. Discussion

Comment 10

References

The authors must format the References according to the journal's instructions

(References should be described as follows, depending on the type of work:

Journal Articles:

1. Author 1, A.B.; Author 2, C.D. Title of the article. Abbreviated Journal Name Year, Volume, page range.)

Increase the number of the reference papers including (primarily) from MDPI Journals. 

The authors use 0 papers from Coatings journal / 2 MDPI Journals / 44 papers from journals (References). 

The number for papers from MDPI journals is considered insufficient (in reviewer's opinion).

Round 2

Reviewer 1 Report

The revised version can be published if other reviewers have no more additional comments. 

Author Response

Once again, we appreciate your insightful comments, which have helped us to enhance the quality of our manuscript. We hope that the revised version of the paper now meets your expectations and that you will find it suitable for publication.

Thank you for your time and effort in reviewing our manuscript. If you have any further comments or suggestions, please do not hesitate to let us know.

Best regards,

Authors

Reviewer 2 Report

Comment 1

Lines 2-3

Characterization and Corrosion Behavior of Zinc Coatings for 2 

Anti-Corrosive Protection: A Detailed Study

The authors should check if the number 2 must be insert in the title.

Line 118

Guo evaluated the influence

The authors should replace

Guo et al. evaluated the influence

Line 942

The ref "Nie, B., Xu, S., Zhang, Z., & Li, A. Surface morphology characteristics and mechanical properties of corroded cold-formed steel channel 

sections. Journal of Building Engineering, 2021, 42, 102786." must remove and renumber (49?).
